# Universal Set Transformer: A Scalable and Interpretable Set/Multiset Architecture

## Abstract

The advent of the set transformer (ST) brought about a new method of permutation equivariant modeling by leveraging cross-element interactions. However, ST is still subject to the fundamental challenge of transformers: scaling efficiently with large input sizes. Mini-batch consistent (MBC) methods were developed to address this problem by maintaining permutation equivariance while alleviating context fragmentation when processing partitioned sets. However, current MBC methods limit expressiveness and render the models incapable of producing element-wise contextualized representations and attention scores for prediction explainability. Therefore, the choice between ST or MBC methods results in a tradeoff between expressiveness and large set processing. To reconcile this tradeoff we propose the Universal Set Transformer (UST), a generalization of ST which is mini-batch consistent without sacrificing expressiveness. Additionally, we introduce multiset attention which leverages the MBC property to significantly reduce the computational cost of processing multisets while maintaining mathematical equivalence with standard multi-head attention. We show that UST is competitive with ST's performance while using less memory and outperforms other MBC methods in various benchmark tasks. Finally, we show that UST is capable of producing both whole-set and element-wise representations and demonstrate prediction explainability via attention scores.

## 1 Introduction

Modeling unstructured data with deep learning poses challenges as most common architectures are designed for ordered data. Extending deep learning to unordered data naively requires training on all possible permutations of the input which has complexity $O(n!)$. Pooling methods like Deep Sets (Zaheer et al., 2017) and FSPool (Zhang et al., 2020) circumvent this intractable complexity by processing each element of the set independently, aggregating the results with an order-agnostic pooling mechanism, and then passing the aggregated result downstream, yielding a complexity of $O(n)$. However, by processing each element independently, the model is ignorant of higher order interactions between elements in the set and how each element behaves in context of the whole set.

Set transformer (ST) (Lee et al., 2019) improved on this idea by recognizing that transformers are permutation equivariant without positional encodings and exploit the transformer's multi-headed self-attention to capture the higher order interactions between set elements that pooling-based approaches lack. However, self-attention also imposes a quadratic computational complexity of $O(n^2)$. ST attempts to mitigate this by proposing an approximation to the full self-attention through induced attention (Lee et al., 2019). Although this reduces the complexity to $O(kn)$, ST still requires access to the full set at once, which may still not fit into memory becoming intractable.

One possible solution to the large set size problem is to shard the set, processing each shard individually and then aggregating the result together. However, this breaks permutation-equivariance in transformer-based architectures and leads to context fragmentation as each partition is processed independently without awareness of the rest of the set which can cause key interactions to be missed. Set Transformer XL (STrXL) by Givens et al. (2024) attempts to mitigate this through the use of a memory mechanism while approximating permutation-equivariance. Rabe & Staats (2022) and Jang et al. (2019) proposed a modification to attention that solves the context-fragmentation by allowing permutation-invariant processing of the input in shards without affecting the result. Mini-batch con-

sistent (MBC) methods for transformer-based architectures were proposed by Andreis et al. (2021) and Willette et al. (2023) to enable permutation-invariant processing of a sharded set leveraging this modification. However, they incur the cost of expressivity as these models can only produce set-summary representations. Furthermore, these methods limit the natural interpretability of self-attention scores that can be used to identify factors that drive model predictions.

Recent examples where contextualized element representations and element-wise attention interpretability are needed is in the field of bioinformatics (Cui et al., 2025; Ruffolo et al., 2021; Ludwig II & Phillips, 2025; Ludwig et al., 2025). The recent SetBERT next-generation sequencing foundation model proposed by Ludwig II & Phillips (2025) relies on subsampling (i.e. using only a portion of the DNA sequences) in order to achieve these goals. Thus, to our knowledge, there is no known architecture capable of meeting all of these requirements.

In this work, we demonstrate the limitations and tradeoffs of each set modeling approach described above, and propose a unified architecture that fully addresses all of these issues, which we call Universal Set Transformer (UST). UST introduces the U-ISAB, a highly-scalable ST-based architecture that achieves MBC processing without sacrificing performance or expressivity. Additionally, UST also introduces Multiset Attention (MSA) that leverages the MBC property for efficient multiset processing that is mathematically equivalent to standard multi-head attention (MHA) with a reduced computational footprint. We demonstrate across a diverse set of tasks that UST represents the new state-of-the-art transformer architecture for MBC processing.

## 2 BACKGROUND

### 2.1 SET TRANSFORMERS

The transformer by Vaswani et al. (2017) is the state-of-the-art deep learning architecture in nearly all domains. The authors of Set Transformer (Lee et al., 2019) recognized that by simply removing position encodings, the transformer becomes a powerful permutation-equivariant architecture that naturally handles unordered data. Unlike prior pooling methods (Zaheer et al., 2017; Willette et al., 2023; Zhang et al., 2020), ST is able to capture higher-order interactions between set elements.

The ST framework defines several components: the *set attention block* (SAB), the *induced set attention block* (ISAB) and *pooling by multi-head attention* (PMA). The SAB is a simplified version of the standard self-attention transformer block by Vaswani et al. (2017). However, the authors realized that the $O(n^2)$ complexity of the SAB severely limited the scalability of the model and proposed the ISAB as a solution. This block uses *induced attention* (also known as *slot-attention*) to approximate full self-attention by using a fixed set of learned inducing points to function as a bottleneck, reducing the complexity down to $O(kn)$ where $k$ is a tuneable hyperparameter. Finally, they leverage the induced attention idea to define the PMA operation, a permutation-invariant pooling strategy utilizing multi-head attention to obtain a fixed size set representation. While the ISAB can handle larger sets, fragmented processing may still be required resulting in context-fragmentation effects (Andreis et al., 2021; Willette et al., 2023; Givens et al., 2024).

### 2.1.1 MINI-BATCH CONSISTENT METHODS

Andreis et al. (2021) and Willette et al. (2023) proposed mini-batch consistent (MBC) methods for transformers as a solution to context fragmentation for sharded processing. MBC processing guarantees that splitting a set into random shards (i.e. mini-batches) and processing each mini-batch separately yields exactly the same result and gradients as processing the full set at once Andreis et al. (2021). Concretely, given a set $X$ split into shards $X_1, ..., X_N$, a function $f$ is MBC if there exists a permutation-invariant aggregator $g$ (e.g. sum, mean, or max) such that $f(X) = g(f(X_1), f(X_2), ..., f(X_n))$.

The Slot Set Encoder (SSE) by Andreis et al. (2021) is a transformer-based architecture that is MBC. The SSE is roughly equivalent to the PMA block from ST. However, the authors recognized that by replacing softmax with sigmoid within the attention function, the block naturally becomes MBC. SSEs can be stacked to create the Hierarchical SSE to model more complex sets. but the lack of element-wise interactions result in poor performance (Willette et al., 2023).

Table 1: Feature comparisons between models and frameworks.

| Model | MBC | Attention | | Representations | | Interpretable |
|---|---|---|---|---|---|---|
| | | Cross | Self | Set | Element | |
| Deep Sets (Zaheer et al. (2017)) | ✓ | ✗ | ✗ | ✓ | ✗ | ✗ |
| FSPool (Zhang et al. (2020)) | ✗ | ✗ | ✗ | ✓ | ✗ | ✗ |
| Set Transformer (Lee et al. (2019)) | ✗ | ✓ | ✓ | ✓ | ✓ | ✓ |
| SSE (Andreis et al. (2021)) | ✓ | ✓ | ✗ | ✓ | ✗ | ✗ |
| UMBC (Willette et al. (2023)) | ✓ | ✓ | ✓ | ✓ | ✗ | ✗ |
| STrXL (Givens et al. (2024)) | ✗ | ✓ | ✓ | ✓ | ✓ | ✓ |
| **UST (Ours)** | ✓ | ✓ | ✓ | ✓ | ✓ | ✓ |

Universal MBC (UMBC) was proposed by Willette et al. (2023) to address the issues of SSE. The authors realized that a single SSE can be used to first pool the input set into a fixed size representation of $k$ elements in an MBC manner, and then process the rest of the set with non-MBC ST blocks. This combination of the SSE and ST architecture enables indirect modeling of element interactions resulting in a notable improvement. Additionally, the authors also recognized that the softmax function can be decomposed to be made MBC, alleviating the need for sigmoid attention. While UMBC can truly scale to sets of any size, it comes with a significant restriction in expressivity as UMBC can only produce set-summary representations.

### 2.2 ON THE INTERPRETABILITY OF SET TRANSFORMER-BASED METHODS

Unlike most deep-learning architectures that are effectively 'black boxes', it has been demonstrated that the attention scores computed within transformers can be mined to explain predictions (Hao et al., 2021; Dosovitskiy et al., 2021; Abnar & Zuidema, 2020; Vig & Belinkov, 2019). Self-Attention Attribution by Hao et al. (2021) has been demonstrated with ST to identify microorganisms that drive predictions from DNA samples (Ludwig et al., 2025). While theoretically compatible with nearly all transformer-based architectures, the reliance on element-wise interactions in these methods suggest that the interpretability of current MBC methods would be inherently limited since the hierarchical SSE and UMBC models never compute cross-element interactions.

## 3 UNIVERSAL SET TRANSFORMER

We now present Universal Set Transformer (UST), an extension of the ST framework that provides MBC set processing for high-scalability without the loss of expressivity or attention interpretability. Table 1 compares the features between UST and other architectures/frameworks.

### 3.1 MULTISET ATTENTION

Multisets are a special case of sets where duplicate elements are present. Standard scaled dot-product attention requires operating over the dense multiset obtained by repeating each element $m_i$ times. Since scaled dot-product attention is a set-equivariant operation (Zhang et al., 2022b), these duplicate elements are transformed identically, resulting in many redundant calculations. This is different than multiset-equivariant methods such as (Zhang et al., 2022b) or Zhang et al. (2023) that aim to transform duplicate elements non-identically. When set-equivariance is desired, we can significantly reduce memory and computation when many duplicates are present by representing the dense set as a sparse multiset via a set of unique elements with corresponding multiplicities. This processing method results in very little overhead, even when few-to-no duplicates are present.

We propose *Multiset Attention*, a modification to scaled dot-product attention (Vaswani et al., 2017) that incorporates per-element key multiplicities $\boldsymbol{m}_k \in \mathbb{N}_0^{n_k}$ directly into the softmax weighting:

$$\text{MultisetAttention}(Q, K, V, \boldsymbol{m}_k) = \text{softmax}\left(\frac{QK^\top}{\sqrt{d_k}} \oplus \log(\boldsymbol{m}_k^\top)\right)V \tag{1}$$

where '$\oplus$' denotes row-wise addition of the vector $\log(\boldsymbol{m}_k^\top)$. The key insight is that adding $\log(m_i)$ inside the softmax is algebraically equivalent to replicating the $i$-th key $m_i$ times in the dense multiset (shown in Equation A.1). For a multiset of size $n_k$ with multiplicities $\{m_i\}$, Multiset Attention

reduces the computational complexity of full self-attention from $O((\sum_i m_i)^2)$ down to $O(n_k^2)$. We demonstrate memory utilization of Multiset Attention in Figure A.1. Furthermore, this formulation naturally handles zero-multiplicity elements as attention masks. Since $\log(0) = -\infty$, any key with $m_i = 0$ contributes $-\infty$ to its softmax input and thus receives zero weight in the attention output. Likewise, large multiplicity values grow logarithmically, making it numerically stable with out-of-the-box softmax implementations in frameworks like PyTorch and Tensorflow.

Lastly, Multiset Attention requires either MBC or full-set processing. This is due to the fact that in dense multiset scenarios, duplicate elements are scattered across mini-batches, so each mini-batch approximates the overall distribution of the set. This allows non-MBC models to often overcome context-fragmentation effects (Ludwig II & Phillips, 2025; Givens et al., 2024; Andreis et al., 2021; Willette et al., 2023). However, the sparse multiset representation inherently assigns all duplicates to the same mini-batch, resulting in each mini-batch no longer resembling the global distribution. Therefore, MBC or full-set processing is required to guarantee reliable performance.

## 3.2 MINI-BATCH CONSISTENT MULTISET ATTENTION

As mentioned previously, Willette et al. (2023) realized that the softmax function in traditional scaled dot-product attention can be decomposed to enable MBC processing. We can therefore incorporate this same decomposition trick into our Multiset Attention function to make it MBC. Given a set of queries $Q \in \mathbb{R}^{n_q \times d_k}$ and a set of sharded keys $K = \{K^{(1)}, ..., K^{(n)}\}$ with corresponding values $V = \{V^{(1)}, ..., V^{(n)}\}$ and key multiplicities $\boldsymbol{m} = \{\boldsymbol{m}^{(1)}, ..., \boldsymbol{m}^{(n)}\}$ where $K^{(p)} \in \mathbb{R}^{n_p \times d_k}$, $V^{(p)} \in \mathbb{R}^{n_p \times d_v}$, and $\boldsymbol{m}^{(p)} \in \mathbb{N}_0^{n_p}$, we redefine Multiset Attention as[1]:

$$\text{MultisetAttention}(Q, K, V, \boldsymbol{m}) = \frac{\sum_p A^{(p)} V^{(p)}}{\sum_p \sum_j A_{\cdot, j}^{(p)}} \in \mathbb{R}^{n_q \times d_v}$$

$$\text{where } A^{(p)} = \exp\left(\frac{QK^{(p)}}{\sqrt{d_k}} \oplus \log \boldsymbol{m}^{(p)}\right) \in \mathbb{R}^{n_q \times n_p}. \tag{2}$$

This can be adapted to multiple attention heads with learned projections $W_i^Q, W_i^K, \in \mathbb{R}^{d_{\text{model}} \times d_k}$, $W_i^V \in \mathbb{R}^{d_{\text{model}} \times d_v}$, and $W^O \in \mathbb{R}^{h d_v \times d_{\text{model}}}$ aligning with standard multi-head attention:

$$\text{MultiHeadMSA}(Q, K, V, \boldsymbol{m}) = \text{Concat}(\text{head}_1, ..., \text{head}_h) W^O$$

$$\text{where } \text{head}_i = \text{MultisetAttention}(QW_i^Q, KW_i^K, VW_i^V, \boldsymbol{m}) \tag{3}$$

## 3.3 MINI-BATCH CONSISTENT SET ATTENTION BLOCKS

The primary limitation of current MBC methods is the reliance on pooling the input set which results in no cross-element interactions and the inability to produce transformed element representations. In slot-attention, cross-attention is computed between the slots (queries) and the input set (keys and values). However, we recognize that since the queries are computed independently, the original input set can be sharded and used as the queries directly, resulting in no set pooling. That is:

$$\text{Concat}(\text{MSA}(Q^{(i)}, K, V, \boldsymbol{m}), \text{MSA}(Q^{(j)}, K, V, \boldsymbol{m})) = \text{MSA}(\text{Concat}(Q^{(i)}, Q^{(j)}), K, V, \boldsymbol{m}) \tag{4}$$

where MSA is `MultiHeadMSA`. With this realization, we can define transformer blocks capable of computing full self attention in a manner that is MBC. We start by defining our primary transformer block, the *Multiset Attention Block*, following the modern standard transformer block implementation which incorporates pre-layer normalization (Xiong et al., 2020):

$$\text{MSAB}(X, Y, \mathbf{m}_Y) = H + \text{FFN}(\text{LayerNorm}(H))$$

$$\text{where } H = X + \text{MultiHeadMSA}(\text{LayerNorm}(X), Y_{\text{ln}}, Y_{\text{ln}}, \mathbf{m}_Y) \tag{5}$$

$$\text{and } Y_{\text{ln}} = \text{LayerNorm}(Y)$$

where `LayerNorm` is layer normalization (Ba et al., 2016) and `FFN` is a two-layer MLP with a ReLU activation. It is important to note that ST and SSE/UMBC both deviate from the typical transformer implementation (Vaswani et al., 2017) and use a row-wise linear projection instead of

---

[1]We provide a numerically-stable algorithm for computing MBC Multiset Attention in Algorithm A.1.

the FFN. While Zhang et al. (2022a) has shown that pre-layer normalization for set transformers performs better, we provide a brief ablation study in Section A.2 covering the performance impact when incorporating pre-layer normalization and the FFN.

We can now leverage the same induced attention idea proposed by ST to derive the *Universal Induced Set Attention Block* (U-ISAB) with $O(nk)$ computational complexity:

$$\text{U-ISAB}_k(X, \mathbf{m}) = \{Y^{(1)}, ..., Y^{(p)}\} \in \mathbb{R}^{n \times d_{\text{model}}}$$
$$\text{where } Y^{(i)} = \text{MSAB}(X^{(i)}, \text{MSAB}(I, X, \mathbf{m}), \mathbf{1}) \in \mathbb{R}^{n_p \times d_{\text{model}}} \tag{6}$$

Using U-ISABs results in an architecture that is scalable to large set sizes while remaining immune to context fragmentation and compatible with existing attention explainability methodologies.

## 4 EXPERIMENTS

We evaluate and compare ST, STrXL, UMBC, and UST across a variety of experiments targeting different capabilities of the architectures. The full data processing and training/evaluation details for experiment are provided in Section A.3. For each experiment, we independently train each model architecture in two settings: full set access, and sharded set access via mini-batching. Models trained with the full set also have the full gradient. Models trained with mini-batching only receive gradients from the first mini-batch. STrXL is an exception since it has its own gradient stopping method. Non-MBC models trained with mini-batching average the pooled representations immediately after the PMA layers. After training a model, the parameters that resulted in the lowest validation loss are chosen for benchmarking. 10 independent models for each configuration are trained and evaluated in order to obtain 95% confidence intervals. The UMBC models are trained with their full unbiased gradient correction scheme (Willette et al., 2023). This scheme is theoretically incompatible with the UST architecture; thus, we resort to solely using `StopGrad` to stop gradient flow of subsequent mini-batches as proposed by Andreis et al. (2021).

In order to provide fair comparisons between the different model concepts, we apply the same pre-layer normalization technique to all models and use an FFN in place of the rFF layers. This ensures that the differences we observe are a result of the fundamental designs. We provide a brief ablation experiment in Section A.2 which demonstrates a significant improvement in all architectures with these modifications. Furthermore, since STrXL has an additional memory length hyperparameter, we set it to be enough to contain one full mini-batch unless stated otherwise. Lastly, we note that ST (ISAB), STrXL, and UST are equivalent architectures when operating on the full sets.

### 4.1 MNIST POINT CLOUD CLASSIFICATION

We train and evaluate 2D point cloud classification using the MNIST dataset (Li Deng, 2012). Each point cloud consists of points randomly sampled from the pixels of the MNIST digits. Each model is trained on point clouds of fixed cardinality consisting of 512 points and evaluated on point clouds consisting of 1024 points. We also analyze training and evaluating on variable-cardinality sets in Section A.3.1. After training, we evaluated each model varying the mini-batch size. We plot the results of models trained on the full set and 8-point mini-batches in Figure 1 and summarize the extrapolated accuracies in Table 2. We find that ST, STrXL, and UST are all comparable in the full-set training context where each model obtained a high degree of accuracy. UMBC's compressed representation limits the information that can be stored and underperforms as a result. Moreover, the non-MBC models can only perform well when enough points are provided simultaneously in a given mini-batch as shown in Figure 1. Mini-batches containing few points cause non-MBC models to suffer from context fragmentation and are unable to effectively leverage neighbor interactions to make reliable predictions.

When trained on mini-batches, the non-MBC models do not reach the same level of accuracy as training on the full set. This is due to the combination of context fragmentation, the very small mini-batch size, and limited gradients. Furthermore, the non-MBC models degrade in performance when extrapolating beyond the 8 pt mini-batches they were trained on. This is likely due to the averaging of the pooled mini-batch representations not being representative of the digit. The MBC models do not obtain the same level of performance as training on the full set because we only allow

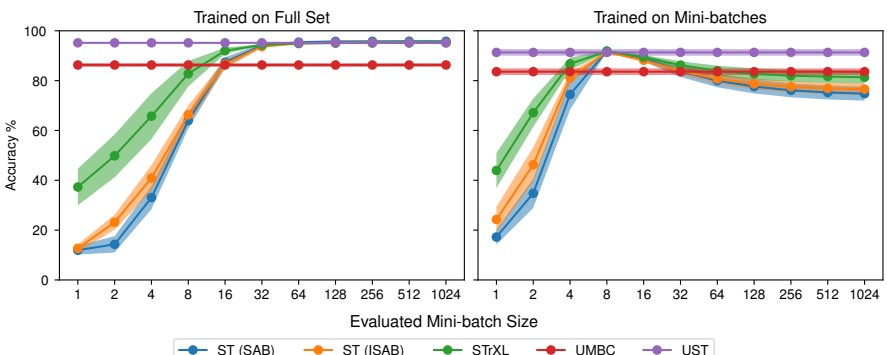

Figure 1: Test accuracy by model for the MNIST point cloud classification task averaged across 10 independent runs with different mini-batch sizes. Each model is evaluated on point clouds consisting of 1024 points. The left plot corresponds to models that had access to the full set (and gradient) during training. The right plot corresponds to models trained with 8-point mini-batches with gradients only computed for one mini-batch. The shaded regions represent 95% confidence intervals.

Table 2: MNIST point cloud classification average test accuracy across 10 independent models trained on the full set and 8-point mini-batches with 95% confidence intervals.

| Mini-batch Size | ST (SAB) | ST (ISAB) | STrXL | UMBC | UST (Ours) |
|---|---|---|---|---|---|
| Full Set | **95.81 ± 0.23** | 95.24 ± 0.26 | 95.22 ± 0.26 | 86.32 ± 0.79 | 95.17 ± 0.31 |
| 8 Points | 74.76 ± 2.81 | 76.55 ± 2.35 | 81.37 ± 2.53 | 83.51 ± 1.33 | **91.35 ± 1.35** |

gradients for the first mini-batch. If gradients present in all mini-batches, the total gradients of the MBC models would be equivalent to training on the full set.

Lastly, we examine the sensitivity to the training mini-batch size by reporting the extrapolated test accuracies in Table 3 of MBC models trained with different mini-batch sizes. We observe that UMBC continues to underperform compared to UST in all configurations. Both models improve performance as the mini-batch size increases since more information is captured in the gradients.

Table 3: Mini-batch size sensitivity for training: Average performance across 10 independent runs with 95% confidence intervals for MNIST point cloud classification and mixture of Gaussians tasks.

| | **MNIST Point Cloud Classification (Accuracy %)** | | **Mixture of Gaussians (NLL)** | |
|---|---|---|---|---|
| Mini-batch Size | UMBC | UST | UMBC (MSA) | UST (MSA) |
| 1 | | | **2.465 ± 0.143** | **2.507 ± 0.140** |
| 2 | | | **5.714 ± 1.297** | **6.161 ± 1.183** |
| 4 | | | **4.581 ± 3.431** | **2.873 ± 0.516** |
| 8 | 83.594 ± 1.325 | **91.333 ± 1.328** | 2.114 ± 0.014 | **2.065 ± 0.019** |
| 16 | 84.752 ± 1.492 | **92.330 ± 0.836** | 2.035 ± 0.007 | **2.004 ± 0.006** |
| 32 | 85.529 ± 1.064 | **92.920 ± 0.621** | 2.001 ± 0.004 | **1.972 ± 0.006** |
| 64 | 85.657 ± 1.172 | **93.668 ± 0.545** | 1.998 ± 0.004 | **1.968 ± 0.003** |
| 128 | 85.471 ± 0.906 | **94.221 ± 0.291** | 2.005 ± 0.003 | **1.968 ± 0.005** |
| 256 | 86.164 ± 1.045 | **94.633 ± 0.290** | 2.009 ± 0.008 | **1.968 ± 0.006** |
| 512 | 85.797 ± 1.014 | **95.024 ± 0.280** | 2.002 ± 0.007 | **1.968 ± 0.004** |

## 4.2 MIXTURE OF GAUSSIAN DENSITY-BASED CLUSTERING

We explore multiset attention by evaluating the MBC models on a modified version of the Mixture of Gaussians task presented by Willette et al. (2023). In this task, 1024 points are sampled from a random weighted mixture of four 2D multivariate Gaussian distributions. We utilize the DBSCAN algorithm (Ester et al., 1996) to cluster points that are close-by to represent them as single points with corresponding multiplicities. The models are then tasked with predicting the cluster parameters.

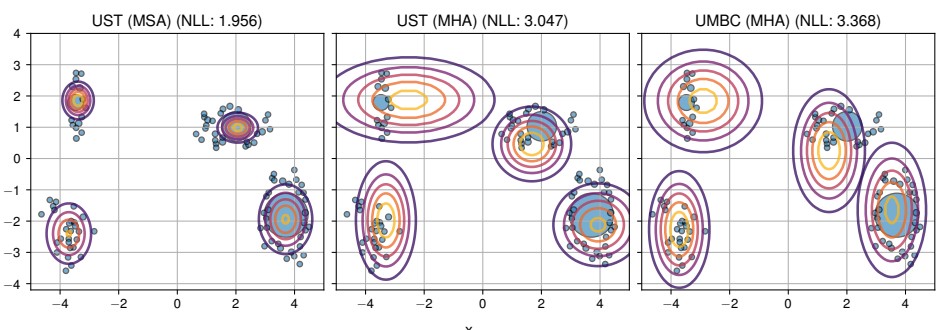

Figure 2: The predicted clusters from UST with multiplicity-weighted multiset attention (MSA) and UMBC using standard muli-head attention (MHA).

Table 4: The mean NLL of each model over 10 runs with 95% confidence intervals.

| UMBC (MHA) | UST (MHA) | UMBC (MSA) | UST (MSA) |
|---|---|---|---|
| 3.514 ± 0.031 | 3.680 ± 0.156 | 2.114 ± 0.014 | **2.065 ± 0.019** |

Willette et al. (2023) has thoroughly demonstrated the failure of non-MBC models to learn this task, so here we focus only on comparing MBC models. We train UST and UMBC on this task for 50 epochs following the same training protocol and model structure as described in the original manuscript (Willette et al., 2023), replacing the standard ST ISAB with ours. The models were provided mini-batches of a fixed number of cluster points each and corresponding multiplicities. Only in the cases where MSA was used was multiplicity factored in. For the standard MHA models, only the representative cluster points were used.

We show the average negative log-likelihood (NLL) resulting from the models trained on 8 cluster points in Table 4. We also visualize the clustering in Figure 2 showcasing the difference between UST with MSA and vanilla UMBC. As expected, the models using standard MHA are not capable of factoring in multiplicity bias into the elements, causing them to underperform. The integration of MSA enables them to significantly improve their abilities to predict the cluster parameters. We further analyze the effect of varying the training mini-batch size with MSA in Table 3. We find that UST and UMBC perform similarly for very small mini-batches since the models only receive gradients for a single point. However, as the mini-batch size increases, UST outperforms UMBC.

### 4.3 Taxonomy Classification via Contextualized Element Representations

In order to demonstrate UST's relevance and scalability in a real-world application, we explore the models' abilities to produce contextualized element representations by predicting genus-level taxa of DNA sequences in context to a microbiome. In this task, the models are presented with a DNA sample from some microbiome consisting of a large number of DNA sequences. In a sample context, the models can leverage information from neighboring DNA sequences in order to more reliably resolve the taxonomies for each DNA sequence. Next generation sequencing technologies such as high-throughput sequencing can yield over 100,000 DNA sequences per sample (Illumina, Inc., 2023). Since the UMBC models rely on an initial pooling of the set, they cannot output element representations and are therefore not applicable to this task. This leaves UST as the only architecture that is both MBC and capable of producing contextualized element representations.

We trained the models for taxonomy classification by drawing 1024 sequences at random from taxonomy profile replicate distributions. Each of the DNA sequences were embedded using DNABERT-S (Zhou et al., 2024). Since each batch is processed i.i.d. for the non-MBC models, and since a full set doesn't exist, we train all models on the 1024-sequence subsamples without any mini-batching. However, we train (and evaluate) STrXL using mini-batches of 512 sequences so that its transformer blocks observe 1024 sequences at a time in order to match the other models. Lastly, because duplicate sequences can naturally occur in this domain, and often make up a signif-

Table 5: The mean taxonomy classification accuracies across 10 runs with 95% confidence intervals.

| Test Size | ST (SAB) | ST (ISAB) | STrXL | UST |
|---|---|---|---|---|
| 1024 | 96.531 ± 0.138 | **97.740 ± 0.100** | **97.735 ± 0.105** | **97.824 ± 0.088** |
| 2048 | 95.995 ± 0.121 | 97.401 ± 0.097 | **97.675 ± 0.081** | **97.762 ± 0.088** |
| 4096 | 95.289 ± 0.094 | 96.760 ± 0.083 | **97.320 ± 0.062** | **97.440 ± 0.064** |
| 8192 | 94.429 ± 0.136 | 96.109 ± 0.088 | 97.061 ± 0.050 | **97.220 ± 0.047** |
| 16384 | 93.242 ± 0.139 | 95.483 ± 0.096 | 96.676 ± 0.049 | **96.845 ± 0.043** |
| 32768 | 91.472 ± 0.168 | 94.600 ± 0.110 | 95.992 ± 0.058 | **96.308 ± 0.033** |
| 65536 | 89.219 ± 0.201 | 93.608 ± 0.127 | 95.268 ± 0.050 | **95.596 ± 0.030** |
| 131072 | 86.427 ± 0.218 | 92.547 ± 0.135 | 94.528 ± 0.051 | **94.769 ± 0.034** |

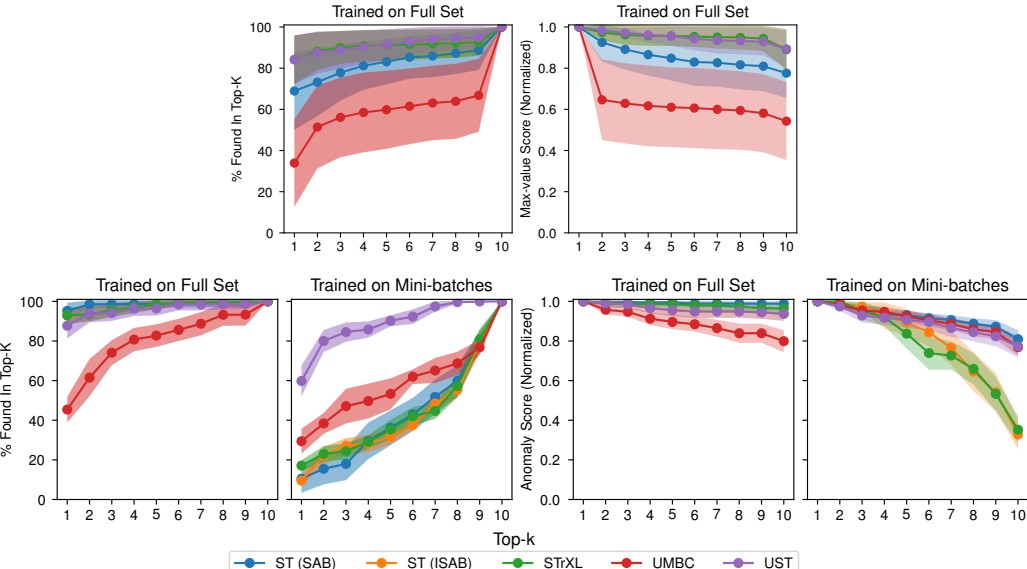

Figure 3: Target input identification accuracies (left) and normalized target scores (right) for max-regression (top) and anomaly detection (bottom) across 10 independent runs. The left plots show the mean percentage of evaluations where the targets were among the top-k highest scores. The right plots show the mean (normalized) score of the targets when identified in the top-k. The shaded regions represent 95% confidence intervals.

icant portion of the reads, we employ MSA in all of the models to improve efficiency by removing duplicates and supplying multiplicity information instead.

We evaluate on increasingly large subsamples comprised of mini-batches consisting of 1024 sequences each. The results are displayed in Table 5. We find that each of the models are able to achieve very high classification accuracy. However, we find that UST and STrXL are able to extrapolate to larger set sizes far more effectively than standard ST. ST SAB achieves the lowest performance, likely having to deal with a lot of noise in the sample as a result of computing full attention. We hypothesize that the induced attention within the ISAB architectures works as a gating mechanism to filter out noise so that more meaningful information is retained when unpacked on the other side. This behavior was also observed by Lee et al. (2019) and Givens et al. (2024).

## 4.4 INTERPRETING PREDICTIONS THROUGH ATTENTION SCORES

In this section, we aim to demonstrate the limitation of attention score interpretability of current MBC methods and show that UST is compatible with existing interpretation methodologies while being MBC. For each of the following experiments, we employ the gradient-based Self-Attention Attribution method by Hao et al. (2021) to score the important input elements.

### 4.4.1 MAXIMUM-VALUE REGRESSION

We demonstrate the interpretability of full set predictions by training the models to regress against the maximum value in a set of 10 integers. The values of each set are chosen from a random uniform distribution in the range of [0, 1000] without any modifications. We found that all but the UMBC models were able learn the task with similar performance as shown in Table 6. The initial pooling of the model appears to severely limit its representation power, and as a result, it cannot produce reasonable predictions. We consistently observed this with other architectural hyperparameters.

We then provided each trained model with 100 sets and mined the attention scores. In Figure 3, we plot the top-k identification accuracies (left) and corresponding mean scores (right). As hypothesized, all but the UMBC models were able to comparatively score the maximum value in the set. The lack of cross-element interactions in UMBC resulted in it being much more unlikely to identify associate the anomalous image with the highest score.

Table 6: The average max-value regression MAE 10 runs with 95% confidence intervals.

| ST (SAB) | ST (ISAB) | STrXL | UMBC | UST |
|----------|-----------|-------|------|-----|
| **5.489 ± 1.754** | **7.076 ± 1.771** | **7.076 ± 1.771** | 159.104 ± 27.674 | **6.972 ± 1.463** |

### 4.4.2 SET ANOMALY DETECTION

We evaluate each model's ability to explain predictions from attention scores using a modified version of the set anomaly task from Lee et al. (2019) using the CelebA dataset (Liu et al., 2015). In this task, a model is presented with a set of 10 images of celebrity faces, each sharing two common attributes (e.g. black hair & mustache) with a 50/50 chance that 1 of the 10 images in the set will be anomalous by sharing neither of these two attributes. The objective of the model is to determine whether or not an anomaly exists in the set (i.e. binary classification). We trained the models with access to the full sets and with mini-batching where each mini-batch contains only one image. The images were pre-embedded using a fine-tuned ViT encoder pre-trained on ImageNet-21k (Deng et al., 2009; Wu et al., 2020). The classification performance results are described in Table 7.

Table 7: The mean set anomaly detection accuracies across 10 runs with 95% confidence intervals.

| Trained On... | ST (SAB) | ST (ISAB) | STrXL | UMBC | UST |
|---------------|----------|-----------|-------|------|-----|
| Full Set | **0.95 ± 0.01** | **0.96 ± 0.01** | **0.96 ± 0.01** | 0.89 ± 0.01 | **0.95 ± 0.01** |
| Mini-batches | 0.57 ± 0.02 | 0.56 ± 0.02 | 0.58 ± 0.01 | **0.81 ± 0.02** | **0.81 ± 0.02** |

We found that all models with access to the full set were able to learn the task with high accuracy. However, UMBC underperformed in comparison to the other models, which we again associate with the limited information in the pooled representation. In the mini-batch setting, the non-MBC models were unable to learn the task due to catastrophic context fragmentation. Even under these extreme conditions, however, the MBC models were still able to learn the task.

After, we again provided each trained model with 100 sets containing an anomalous image and analyzed the attention scores shown in Figure 3. Like the max-value regression task, we observed that all but the UMBC models were able to associate the anomalous image among the highest scores when trained on the full set. In the mini-batched training context, UST was able to significantly outperform all other models in identifying the anomaly through attention analysis.

These results of these attention-interpretability experiments conclude that previous MBC methods (UMBC) are severely limited in attention interpretability compared the other models. When trained with access to the full set, UMBC cannot overcome the lack of cross-element interactions to provide meaningful interpretability. When trained in the mini-batch setting, the non-MBC models are severely hindered by context fragmentation and also cannot leverage cross-element interactions effectively, causing them to perform worse. UMBC performs better than non-MBC models in this case since it is not affected by mini-batching. However, UST significantly outperforms the other models in this scenario since it is able to fully leverage cross-element interactions without context-fragmentation.

# 5 DISCUSSION

In this work, we demonstrated the limits of current ST-based and MBC architectures and proposed Universal Set Transformer to address all of the issues presented. We demonstrated that UST achieves MBC processing while having better performance and more expressive power than previous MBC methods, representing the new state-of-the-art for transformer-based MBC processing. Moreover, we introduced Multiset Attention that leverages the MBC property to process sparse multisets equivalently to dense representations with a reduced computational footprint. Lastly, we showed that previous MBC methods have limited interpretability via attention scores compared to UST and others.

While UST achieves state-of-the-art performance, it is important to note that, unlike UMBC where mini-batches are immediately discarded after use, the U-ISAB must maintain one full instance of the set prior to pooling. However, unlike ST's ISAB, since our U-ISAB is MBC, we can offload the unused intermediate mini-batches to another storage medium (i.e. CPU RAM, disk, etc.). This enables UST to scale far beyond what ST's ISAB is capable of.

Multiset attention is our approach of further reducing both computation and memory when working with multisets. However, it comes at the cost of being strictly limited to set-equivariant processing (Zhang et al., 2022b) and requires knowing element multiplicity information which may not always be known ahead of time. The true multiplicity of multisets can be determined through sorting or hash-based approaches, and the approximate multiplicity sets with similar items can be computed through clustering algorithms such as voxelization, DBSCAN (Ester et al., 1996), etc. One can determine when computing such multiplicities using a particular method is beneficial through complexity analysis. Given the $O(nk)$ complexity the U-ISAB, and the computational complexity of a particular multiset identification/approximation algorithm $O(C_n)$, the algorithm is beneficial when $O(C_n + n(1-r)k) < O(nk) \implies r > \frac{C_n}{nk}$ where $r$ is the redundancy rate of the multiset.

When working with multisets that have few-to-no duplicates present in the set, multiset attention cannot be used to improve memory utilization. Therefore, if MBC processing of UST is not enough fit within memory constraints, it may be reasonable to relax the MBC requirement and opt for non-MBC optimizations. STrXL by Givens et al. (2024), along with the results of our experiments, show that approximately-permutation-equivariant models can still learn and perform well. Therefore, it is reasonable to suggest that incorporating methods such as sparse-attention into UST as an alternative means of reducing memory utilization at the cost of MBC processing would still result in performant models. However, it is worth noting that, due to the complete removal of cross-element interactions in sparse-attention methods, they are not compatible with multiset-attention.

We would like to emphasize the significance of our taxonomy classification experiment. The number of input tokens supplied in one set exceeds the previous largest (to our knowledge) by one order of magnitude (Willette et al., 2023). Also to our knowledge, UST is the only MBC architecture capable of producing contextualized representations, fully addressing the current limitations of recent next-generation sequencing models (Ludwig II & Phillips, 2025; Ludwig et al., 2025).

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

# A    APPENDIX

## A.1    MULTISET ATTENTION

For any logit vector $\mathbf{x}$ of length $n$ and multiplicities $\mathbf{m}$, the following shows the exact equivalence to the dense-multiset formulation without any explicit replication of logits:

$$\text{softmax}_i(\mathbf{x} + \log \mathbf{m}) \;=\; \frac{e^{x_i + \log m_i}}{\sum_j e^{x_j + \log m_j}} \;=\; \frac{m_i e^{x_i}}{\sum_j m_j e^{x_j}} \;=\; \text{softmax}_i(\mathbf{x}, \mathbf{m}) \tag{A.1}$$

### A.1.1    MEMORY EFFICIENCY & SCALABILITY

We show the raw memory usage of multiset attention with vanilla ST's SAB and ISAB blocks in Figure A.1 for a multiset consisting of $2^{32} = 4,194,304$ elements by varying the number of unique items. Peak memory usage is reported as observed from Pytorch. Since we are not mini-batching in this benchmark, our U-ISAB is equivalent to ST's ISAB in terms of memory usage.

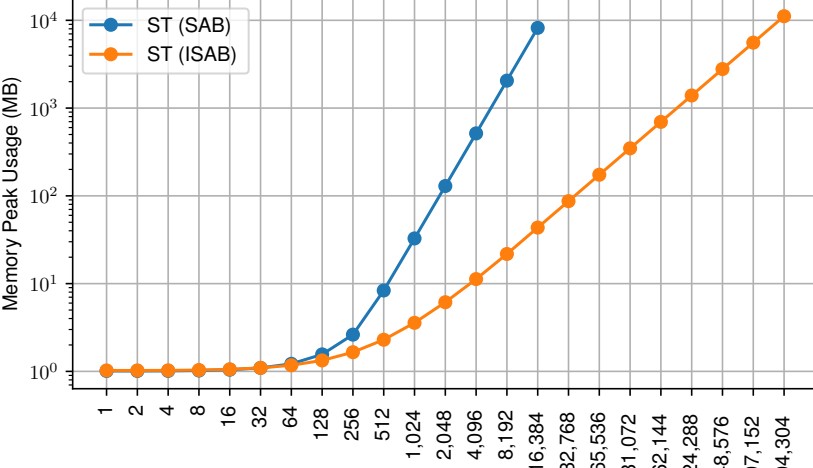

Figure A.1: The peak memory usage for a single SAB and ISAB using multiset attention when processing a fixed size multiset consisting of $2^{32} = 4,194,304$ elements. The number of unique elements is varied from 1 (all duplicates) to $2^{32}$ (no duplicates), incrementing by powers of 2. The SAB and ISAB both have 4 attention heads, $d_{\text{model}} = 32$, and the ISAB has $n_{\text{slots}} = 4$.

### A.1.2    NUMERICALLY-STABLE MBC MULTISET ATTENTION IMPLEMENTATION

Algorithm A.1 below provides a numerically stable implementation of MBC Multiset Attention in PyTorch. This algorithm makes use of the max-shift trick to make softmax numerically stable, and uses an additional down-scaling trick to fit the current mini-batch and running sums into the same frame of reference.

## A.2    TRANSFORMER MODIFICATION ABLATION STUDY

Here we perform a brief ablation study on the incorporation of pre-layer normalization and FFN in the Set Transformer architectures used in this work. We trained each model 10 different times on the MNIST point cloud classification task and evaluated them on the held-out test set. The results are shown it Table A.1 Since STrXL model already incorporates these changes in its architecture, we only show ST, UMBC, and UST.

**Algorithm A.1** Numerically-stable MBC Multiset Attention (PyTorch Implementation).

```python
def mbc_multiset_attention(Q, kvm_mini_batches):
    """
    Q: Tensor[b, n_q, d]
    K: Tensor[b, n_k, d]
    V: Tensor[b, n_k, d]
    m: Tensor[b, n_k]
    """
    global_attention = 0.0
    global_norm = 0.0
    global_max = None
    for (K, V, m) in kvm_mini_batches:
        # Compute multiset attention (numerator)
        log_m = torch.log(m).unsqueeze(1)
        local_attention = Q @ K.transpose(1, 2) / math.sqrt(d) + log_m

        # Scale attention before exponentiating for numerical stability
        local_max = attention.amax(dim=-1, keepdim=True)
        local_attention = torch.exp(local_attention - local_max)

        # Zero out NaNs (i.e. from empty sets) and compute running sum
        local_attention = attention.mask_fill(torch.isnan(attention), 0.0)
        local_norm = attention.sum(dim=-1, keepdim=True)

        # Assign global_max if currently unassigned
        if global_max is None:
            global_max = local_max

        # Scale down each state to fit in the same frame of reference
        delta_global = torch.exp(torch.clamp(global_max - local_max,
        ↪   max=0.0))
        delta_local = torch.exp(torch.clamp(local_max - global_max, max=0.0))
        global_attention = global_attention*delta_global + (local_attention @
        ↪   V)*delta_local
        global_norm = global_norm*delta_global + local_norm*delta_local

        # Update global max
        global_max = torch.maximum(global_max, local_max)

    # Compute the output. Zero out NaNs from empty sets
    output = global_attention / global_norm
    output.masked_fill(output.isnan(), 0.0)
    return output
```

Table A.1: MNIST point cloud classification average test accuracy across 10 independent runs with 95% confidence intervals. The models here were trained with full access to the set and gradient and evaluated on the full 512 pt point clouds. The ablation study compares the effect of using pre-layer normalization and replacing the row-wise feed-forward layer with the standard feed-forward network in the models.

| Model | Vanilla | Pre-Layer Norm | FFN | Pre-Layer Norm + FFN |
|---|---|---|---|---|
| ST (SAB) | $0.938 \pm 0.004$ | $0.939 \pm 0.004$ | $0.941 \pm 0.002$ | $\mathbf{0.955 \pm 0.003}$ |
| ST (ISAB) | $0.932 \pm 0.004$ | $0.928 \pm 0.005$ | $0.937 \pm 0.005$ | $\mathbf{0.949 \pm 0.002}$ |
| UMBC | $0.772 \pm 0.034$ | $\mathbf{0.845 \pm 0.009}$ | $0.760 \pm 0.017$ | $\mathbf{0.856 \pm 0.009}$ |
| UST | $0.932 \pm 0.004$ | $0.928 \pm 0.004$ | $0.937 \pm 0.005$ | $\mathbf{0.948 \pm 0.004}$ |

## A.3 EXPERIMENTS

The following subsections describe the complete architecture, training, and evaluation details for each of the experiments presented in the main manuscript.

For each model configuration, 10 independent models were trained in every experiment. During mini-batched training, gradients were only computed for the first mini-batch. `StopGrad` was used on subsequent mini-batches immediately before pooling. All UMBC models were trained with their proposed unbiased gradient correction technique, while all other models were trained without it. For non-MBC models, the average of the pooled representation resulting from PMA across the mini-batches is used for the downstream layers. For MBC models, we utilize the same tricks to allow PMA to be MBC. Lastly, all models utilized 4 attention heads in each transformer-based component.

### A.3.1 MNIST Point Cloud Classification

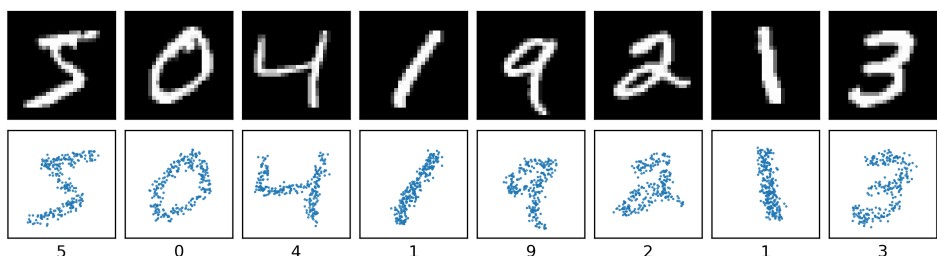

Figure A.2: Example 2D point clouds sampled from MNIST digits.

The MNIST dataset comes pre-split with 60,000 training examples and 10,000 testing examples. Each image example is 28x28 pixels in size. Points were sampled via pixel coordinates following a multinomial distribution weighted by pixel brightness. Random jitter sampled from $\sim \mathcal{N}(0, 0.5)$ for each point and added Finally, the points were normalized roughly to the range of $[-1, 1]$ by dividing each point coordinate by 13.5 and subtracting 1. Examples point clouds are shown in Figure A.2.

The architectural details of the models used in this experiment are shown in Tables A.2 and A.3. Each model was trained for 10 epochs with a batch size of 128 with a random 90/10 train/validation split. Models were trained with fixed cardinalities consisting of 512 points and variable cardinalities sampled uniformly in the range [32-512]. For each configuration, 10 models were trained independently with access to the full set and gradient, and another 10 models were trained independently with mini-batch access to the set and only the gradients from the first mini-batch. For mini-batch training, the input sets were randomly sharded into 8-point mini-batches. The Adam optimizer was used with a constant learning rate of 1e-4.

Table A.2: The MNIST point cloud classification architecture for ST, STrXL, and UST. Set Encoder corresponds to SAB for ST (SAB), $ISAB_8$ for ST (ISAB) and STrXL, and $U\text{-}ISAB_8$ for UST.

| Output Size | Layer | Amount |
|---|---|---|
| $N_i \times 2$ | Input | $\times 1$ |
| $N_i \times 32$ | Linear(2, 32) | $\times 1$ |
| $N_i \times 32$ | Set Encoder | $\times 4$ |
| $1 \times 32$ | $PMA_1$ | $\times 1$ |
| $1 \times 10$ | Linear(32, 10), Softmax | $\times 1$ |

Table A.3: The MNIST point cloud classification architecture for UMBC.

| Output Size | Layer | Amount |
|---|---|---|
| $N_i \times 2$ | Input | $\times 1$ |
| $N_i \times 32$ | Linear(2, 32) | $\times 1$ |
| $8 \times 32$ | $SSE_8$ | $\times 1$ |
| $8 \times 32$ | SAB | $\times 3$ |
| $1 \times 32$ | $PMA_1$ | $\times 1$ |
| $1 \times 10$ | Linear(32, 10), Softmax | $\times 1$ |

After training, each of the models were evaluated on the pre-split test examples. The models trained on fixed-cardinality sets were evaluated with 1,024 points per example, and models trained on variable-cardinality sets were evaluated with examples whose cardinalities were sampled uniformly from [32-1,024]. The expansion of the range to 1,024 points is used to test the models' abilities to extrapolate. All models were evaluated via mini-batches, varying the number of points from 1 to 1,024, incrementing by powers of 2. A mini-batch of 1 is the most extreme scenario where non-MBC models like ST cannot leverage any interactions between neighbors. We plot the fixed-cardinality results for all models in Figure 1 and provide the accuracy values in Tables A.4 and A.5.

Next, we evaluate the variable-cardinality version of task by analyzing the accuracies in Figure A.3. The numeral values are provided in Tables A.6 and A.7. For the models trained on the full set, we find that, with the exception of UMBC, they all perform similarly when given access to the full set at once. However, during mini-batching, the non-MBC models quickly break down as they were not previously required to aggregate mini-batches to form a single prediction. Surprisingly, UMBC seems to perform poorly for this task. When trained on 8-point mini-batches, we again see the same behavior where the models tend to perform their best when the mini-batches consist of 8 points. UST consistently outperforms the other models with the exception of STrXL for small point mini-batches. This is due to the fact that STrXL's memory mechanism provides each mini-batch with additional points, effectively scaling the mini-batch size which improve prediction over other models. However, it does not generalize to all mini-batch sizes and breaks down once the mini-batch size becomes large enough.

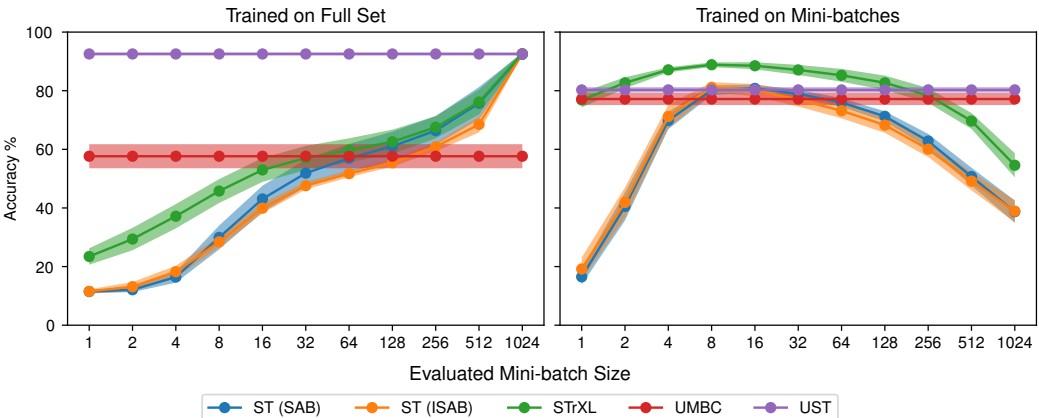

Figure A.3: Test accuracy by model for the MNIST point cloud classification task using variable-cardinality sets averaged across 10 independent runs with different mini-batch sizes. Each model is evaluated on point clouds whose cardinality is sampled uniformly in the range [32-1024]. The left plot corresponds to the models that had access to the full set (and gradient) during training. The right plot corresponds to models trained with 8-point mini-batches with gradients only computed for one mini-batch. The shaded regions represent 95% confidence intervals.

### A.3.2   MIXTURE OF GAUSSIAN DENSITY-BASED CLUSTERING

This task requires the models to predict the parameters of four different 2D multivariate Gaussian distributions. A random categorical distribution is drawn from a Dirichlet distribution to weight which Gaussian to sample points from. Each Gaussian mean is sampled uniformly in the range of $[-4, 4]$ with random variance diagonals in the range $[0.1, 0.6]$. After sampling 1,024 points, we employ the DBSCAN algorithm (Ester et al., 1996) with $\epsilon = 0.2$ to cluster the points that are close together. We then provide the cluster points to the model as multisets with the multiplicities being the number of points that make up the cluster. The models predict five values for each cluster: it's weight, 2D mean, and 2D diagonal variance.

We trained 10 independent models for each configuration on. The model architectures follow the design proposed by Willette et al. (2023) and are shown in Tables A.8 and A.9. The models were trained for 50,000 steps with a batch size of 10 using the Adam optimizer with a scheduled learning

Table A.4: MNIST point cloud classification average test accuracy across 10 independent runs with 95% confidence intervals. The models here were trained with full access to the set and gradient.

| MB Size | ST (SAB) | ST (ISAB) | STrXL | UMBC | UST |
|--------:|----------|-----------|-------|------|-----|
| 1 | 11.93 ± 1.76 | 12.67 ± 1.69 | 37.30 ± 7.36 | 86.32 ± 0.79 | **95.17 ± 0.31** |
| 2 | 14.31 ± 3.30 | 23.16 ± 2.98 | 49.82 ± 8.61 | 86.32 ± 0.79 | **95.17 ± 0.31** |
| 4 | 33.07 ± 4.70 | 40.90 ± 5.20 | 65.73 ± 9.22 | 86.32 ± 0.79 | **95.17 ± 0.31** |
| 8 | 63.96 ± 3.20 | 66.28 ± 3.77 | 82.77 ± 5.11 | 86.32 ± 0.79 | **95.17 ± 0.31** |
| 16 | 87.35 ± 2.07 | 86.69 ± 1.90 | 92.00 ± 1.25 | 86.32 ± 0.79 | **95.17 ± 0.31** |
| 32 | 94.15 ± 0.61 | 93.69 ± 0.54 | 94.37 ± 0.33 | 86.32 ± 0.79 | **95.17 ± 0.31** |
| 64 | **95.45 ± 0.26** | **94.97 ± 0.25** | 94.90 ± 0.24 | 86.32 ± 0.79 | 95.17 ± 0.31 |
| 128 | **95.75 ± 0.22** | 95.23 ± 0.25 | 95.07 ± 0.25 | 86.32 ± 0.79 | 95.17 ± 0.31 |
| 256 | **95.78 ± 0.22** | 95.27 ± 0.25 | 95.13 ± 0.26 | 86.32 ± 0.79 | 95.17 ± 0.31 |
| 512 | **95.81 ± 0.23** | 95.26 ± 0.25 | 95.19 ± 0.24 | 86.32 ± 0.79 | 95.17 ± 0.31 |
| 1024 | **95.81 ± 0.23** | 95.24 ± 0.26 | 95.22 ± 0.26 | 86.32 ± 0.79 | 95.17 ± 0.31 |

Table A.5: MNIST point cloud classification average test accuracy across 10 independent runs with 95% confidence intervals. The models here were trained with mini-batches consisting of 8 points each and access to the gradient of only the first mini-batch.

| MB Size | ST (SAB) | ST (ISAB) | STrXL | UMBC | UST |
|--------:|----------|-----------|-------|------|-----|
| 1 | 17.23 ± 3.07 | 24.27 ± 5.15 | 43.96 ± 7.04 | 83.51 ± 1.33 | **91.35 ± 1.35** |
| 2 | 34.78 ± 5.85 | 46.24 ± 6.98 | 67.13 ± 5.68 | 83.51 ± 1.33 | **91.35 ± 1.35** |
| 4 | 74.46 ± 6.10 | 81.11 ± 2.40 | 86.74 ± 2.50 | 83.51 ± 1.33 | **91.35 ± 1.35** |
| 8 | **91.86 ± 0.35** | **91.40 ± 0.22** | **91.75 ± 0.40** | 83.51 ± 1.33 | 91.35 ± 1.35 |
| 16 | **88.91 ± 1.24** | 88.22 ± 0.79 | **89.40 ± 0.92** | 83.51 ± 1.33 | 91.35 ± 1.35 |
| 32 | 83.87 ± 2.39 | 83.96 ± 1.48 | 86.24 ± 1.53 | 83.51 ± 1.33 | **91.35 ± 1.35** |
| 64 | 80.00 ± 2.66 | 80.69 ± 1.87 | 83.98 ± 2.02 | 83.51 ± 1.33 | **91.35 ± 1.35** |
| 128 | 77.64 ± 2.78 | 78.70 ± 2.15 | 82.66 ± 2.28 | 83.51 ± 1.33 | **91.35 ± 1.35** |
| 256 | 76.11 ± 2.80 | 77.57 ± 2.23 | 81.98 ± 2.40 | 83.51 ± 1.33 | **91.35 ± 1.35** |
| 512 | 75.25 ± 2.80 | 76.92 ± 2.30 | 81.62 ± 2.48 | 83.51 ± 1.33 | **91.35 ± 1.35** |
| 1024 | 74.76 ± 2.81 | 76.55 ± 2.35 | 81.37 ± 2.53 | 83.51 ± 1.33 | **91.35 ± 1.35** |

rate of 1e-3 for the first 35,000 steps and 1e-4 for the last 15,000. The models trained with mini-batching had access to 8 cluster points a time.

### A.3.3 TAXONOMY CLASSIFICATION VIA CONTEXTUALIZED ELEMENT REPRESENTATIONS

We generated 10 de novo community profiles following the gold standard methodology from CAMISIM (Fritz et al., 2019) utilizing DNA sequences and taxonomy labels from the SILVA dataset (Quast et al., 2013). We use the version of SILVA provided by the QIIME 2 platform (Bolyen et al.,

Table A.6: MNIST variable-cardinality point cloud classification average test accuracy across 10 independent runs with 95% confidence intervals. The models here were trained with full access to the set and gradient.

| MB Size | ST (SAB) | ST (ISAB) | STrXL | UMBC | UST |
|--------:|----------|-----------|-------|------|-----|
| 1 | 11.48 ± 0.60 | 11.55 ± 0.70 | 23.44 ± 2.80 | 57.64 ± 4.11 | **92.54 ± 0.52** |
| 2 | 12.10 ± 0.82 | 13.14 ± 1.58 | 29.39 ± 3.83 | 57.64 ± 4.11 | **92.54 ± 0.52** |
| 4 | 16.42 ± 1.91 | 18.24 ± 2.04 | 37.18 ± 4.15 | 57.64 ± 4.11 | **92.54 ± 0.52** |
| 8 | 29.94 ± 4.07 | 28.44 ± 1.87 | 45.77 ± 4.08 | 57.65 ± 4.11 | **92.54 ± 0.52** |
| 16 | 43.07 ± 4.53 | 39.93 ± 1.72 | 52.98 ± 4.08 | 57.65 ± 4.11 | **92.54 ± 0.52** |
| 32 | 51.88 ± 4.69 | 47.60 ± 1.28 | 57.03 ± 4.14 | 57.65 ± 4.11 | **92.54 ± 0.52** |
| 64 | 56.98 ± 4.86 | 51.70 ± 1.23 | 59.66 ± 4.11 | 57.65 ± 4.11 | **92.54 ± 0.52** |
| 128 | 61.05 ± 4.98 | 55.40 ± 1.25 | 62.59 ± 4.12 | 57.65 ± 4.11 | **92.54 ± 0.52** |
| 256 | 66.39 ± 4.96 | 60.89 ± 1.59 | 67.52 ± 3.80 | 57.65 ± 4.11 | **92.54 ± 0.52** |
| 512 | 75.74 ± 5.52 | 68.49 ± 2.73 | 76.13 ± 4.26 | 57.64 ± 4.11 | **92.54 ± 0.52** |
| 1024 | **92.39 ± 0.44** | **92.62 ± 0.56** | **92.59 ± 0.61** | 57.65 ± 4.11 | **92.54 ± 0.52** |

Table A.7: MNIST variable-cardinality point cloud classification average test accuracy across 10 independent runs with 95% confidence intervals. The models here were trained with mini-batches and access to the gradient of only the first mini-batch.

| MB Size | ST (SAB) | ST (ISAB) | STrXL | UMBC | UST |
|---|---|---|---|---|---|
| 1 | 16.50 ± 2.69 | 19.17 ± 3.89 | **76.85 ± 2.55** | 77.16 ± 2.06 | **80.28 ± 0.94** |
| 2 | 40.52 ± 4.78 | 41.88 ± 5.16 | **82.68 ± 1.92** | 77.16 ± 2.06 | **80.28 ± 0.94** |
| 4 | 69.85 ± 2.98 | 71.20 ± 3.97 | **87.13 ± 0.87** | 77.16 ± 2.06 | 80.28 ± 0.94 |
| 8 | 80.17 ± 1.67 | 81.12 ± 1.77 | **88.86 ± 0.81** | 77.16 ± 2.06 | 80.28 ± 0.94 |
| 16 | 80.79 ± 1.32 | 80.21 ± 2.08 | **88.51 ± 1.22** | 77.16 ± 2.06 | 80.28 ± 0.94 |
| 32 | 78.93 ± 1.39 | 76.97 ± 2.38 | **87.09 ± 1.72** | 77.16 ± 2.06 | 80.28 ± 0.94 |
| 64 | 75.97 ± 1.49 | 73.17 ± 2.59 | **85.26 ± 2.19** | 77.16 ± 2.06 | 80.28 ± 0.94 |
| 128 | 71.26 ± 1.72 | 68.27 ± 2.64 | **82.70 ± 2.45** | 77.16 ± 2.06 | **80.28 ± 0.94** |
| 256 | 62.83 ± 2.44 | 60.06 ± 2.68 | **78.40 ± 2.53** | 77.16 ± 2.06 | **80.28 ± 0.94** |
| 512 | 50.70 ± 3.04 | 49.14 ± 3.11 | 69.68 ± 2.71 | 77.16 ± 2.06 | **80.28 ± 0.94** |
| 1024 | 38.72 ± 3.82 | 38.91 ± 3.76 | 54.57 ± 4.12 | 77.16 ± 2.06 | **80.28 ± 0.94** |

Table A.8: The mixture of Gaussians model architecture for UST.

| Output Size | Layer | Amount |
|---|---|---|
| $N_i \times 2$ | Input | ×1 |
| $N_i \times 128$ | Linear(2, 128), ReLU | ×1 |
| $N_i \times 128$ | Linear(128, 128) | ×1 |
| $N_i \times 128$ | U-ISAB$_4$ | ×1 |
| $4 \times 128$ | PMA$_4$ | ×1 |
| $4 \times 128$ | SAB | ×3 |
| $4 \times 5$ | Linear(128, 5) | ×1 |

2019) where the sequences have been trimmed to the hypervariable V3-V4 region of 16S rRNA. For this task, we truncated the taxonomy labels to the genus level. This dataset contains 313,734 unique DNA sequences and 8,415 unique taxonomy labels in this configuration. Each base community profile was sampled from a random log-normal distribution with $\mu = 1.0$ and $\sigma = 2.0$. Then, for each base profile, we produced 100 training and 10 testing replicates by adding relative Gaussian noise. Noise was sampled for each replicate via $\sim \mathcal{N}(0, \sigma_i)$, where $\sigma_i$ is sampled uniformly from the range $[0.5, 0.8]$ for the $i$th base profile.

We pre-embedded all of the DNA sequences without truncation using the official pre-trained DNABERT-S ( Zhou et al. (2024)) model. This model embeds sequences of any length into 768D representations. Each DNA sequence embedding was obtained by computing the mean of all of the transformed token representations (Zhou et al., 2024).

Because samples are generated on the fly following the taxonomy distributions, there is no concept of a full set. Thus, we trained the models on random fixed-size samples consisting of 1,024 DNA sequence embeddings per sample without mini-batching generated from the training replicates. A batch of sequences is produced by randomly sampling taxa and corresponding DNA sequences according to a replicate distribution. This forms a contextualized sample. A learnable class token embedding is also appended to the sample before passing through the model in order capture sample-level information (Wu et al., 2020; Devlin et al., 2019).

10 independent models were trained for each configuration for 50 epochs with a batch size of 8 using the Adam optimizer with a fixed learning rate of 1e-4. The model details are shown in Table A.10.

Each model was evaluated on the test replicates in mini-batches consisting of 1,024 DNA sequences. The number of mini-batches per sample was varied by powers of 2. The corresponding accuracies and confidence intervals are shown in Table A.11.

Table A.9: The mixture of Gaussians model architecture for UMBC.

| Output Size | Layer | Amount |
|---|---|---|
| $N_i \times 2$ | Input | $\times 1$ |
| $N_i \times 128$ | Linear(2, 128), ReLU | $\times 1$ |
| $N_i \times 128$ | Linear(128, 128) | $\times 1$ |
| $4 \times 128$ | $SSE_4$ | $\times 1$ |
| $4 \times 128$ | SAB | $\times 3$ |
| $4 \times 5$ | Linear(128, 5) | $\times 1$ |

Table A.10: The taxonomy classification architecture for ST, STrXL, and UST. Set Encoder corresponds to SAB for ST (SAB), $ISAB_{16}$ for ST (ISAB) and STrXL, and U-$ISAB_{16}$ for UST.

| Output Size | Layer | Amount |
|---|---|---|
| $N_i \times 768$ | Input | $\times 1$ |
| $N_i \times 256$ | Linear(768, 256) | $\times 1$ |
| $N_i \times 256$ | Set Encoder | $\times 4$ |
| $N_i \times N_l$ | Linear(256, $N_l$), Softmax | $\times 1$ |

### A.3.4 MAXIMUM VALUE REGRESSION

For this experiment, we trained the models to predict the maximum value in a set of $l = 10$ integers. For a given set $X$, each element $x_i$ is drawn i.i.d. from $\text{Uniform}(0, v_{\max})$, where the upper bound itself is randomly chosen as $v_{\max} \sim \text{Uniform}(0, 1000)$. 10 independent models were trained for each configuration for 50 epochs with a batch size of 32, where each epoch consists of 1,000 random sets. Loss was computed with mean-squaed error and the Adam optimizer was used with a constant learning rate of $1e - 3$. The architectures for the models are shown in Tables A.12 and A.13.

The models were evaluated using a fixed testing set of 100 sets of 10 integers. Regression performance was computed for each model across all sets using mean-squared error. For prediction explainability, attention attribution scores were computed across the test sets with the *self-attention attribution* (Hao et al., 2021) method using 20 integration steps on all of the transformer layers. The models were provided the full sets for this analysis. For ISAB layers, we only utilized the first inner MAB/MSAB layer to compute the scores. The attribution scores were clipped to the range of $[0, \infty)$ and then aggregated across heads, layers, and rows via summation. Finally, the scores were normalized to the range of $[0, 1]$. This yields a single vector for a set of inputs where the value of each element indicates the importance of the corresponding input element.

### A.3.5 SET ANOMALY DETECTION

This experiment utilized the CelebA dataset. This dataset consists of 202,599 cropped and aligned images of celebrities. Each image has a corresponding binary feature vector indicating the presence or absence of 40 different possible features. The CelebA dataset comes pre-split into three partitions: training, validation, and testing. We only utilized the training and testing splits in this work.

We pre-embedded all of the CelebA images for this task using a fine-tuned a vision transformer (ViT) (Dosovitskiy et al., 2021) pre-trained on ImageNet-21k (Deng et al., 2009). This model and its image processor was pulled directly from Huggingface via `google/vit-base-patch16-224-in21k`. We fine-tuned this model by predicting the binary features of the images using the training split of the data. It was trained for 10 epochs with a batch size of 128 using the Adam optimizer with a fixed learning rate of 1e-4. The images were scaled to 224x224 and pixel values were normalized to the range of $[-1, 1]$.

10 independent models for each configuration were trained on this task to predict the binary presence of an anomaly image in a set. The architectures are described in Tables A.14 and A.15. Each set consists of 10 images drawn uniformly, all sharing two common features. However, there is a 50% chance that one of the images shares neither of the two common features. In this case, the set should be classified as anomalous. Figure A.4 provides some example anomalous sets. The models trained

Table A.11: The mean taxonomy classification accuracies across 10 runs with 95% confidence intervals.

| Test Size | ST (SAB) | ST (ISAB) | STrXL | UST |
|---|---|---|---|---|
| 1024 | 96.531 ± 0.138 | **97.740 ± 0.100** | **97.735 ± 0.105** | **97.824 ± 0.088** |
| 2048 | 95.995 ± 0.121 | 97.401 ± 0.097 | **97.675 ± 0.081** | **97.762 ± 0.088** |
| 4096 | 95.289 ± 0.094 | 96.760 ± 0.083 | **97.320 ± 0.062** | **97.440 ± 0.064** |
| 8192 | 94.429 ± 0.136 | 96.109 ± 0.088 | 97.061 ± 0.050 | **97.220 ± 0.047** |
| 16384 | 93.242 ± 0.139 | 95.483 ± 0.096 | 96.676 ± 0.049 | **96.845 ± 0.043** |
| 32768 | 91.472 ± 0.168 | 94.600 ± 0.110 | 95.992 ± 0.058 | **96.308 ± 0.033** |
| 65536 | 89.219 ± 0.201 | 93.608 ± 0.127 | 95.268 ± 0.050 | **95.596 ± 0.030** |
| 131072 | 86.427 ± 0.218 | 92.547 ± 0.135 | 94.528 ± 0.051 | **94.769 ± 0.034** |

Table A.12: The maximum-value regression architecture for UST.

| Output Size | Layer | Amount |
|---|---|---|
| $N_i \times 1$ | Input | $\times 1$ |
| $N_i \times 64$ | Linear(1, 64) | $\times 1$ |
| $N_i \times 64$ | U-ISAB$_4$ | $\times 2$ |
| $1 \times 64$ | PMA$_1$ | $\times 1$ |
| $1 \times 1$ | Linear(64, 1) | $\times 1$ |

on the pre-embedded images for 7,500,000 steps with a batch size of 128 using the Adam optimizer with a constant learning rate of $1e - 3$.

The models were evaluated using the pre-embedded testing split of the CelebA dataset. Classification accuracy was computed for each model across 5,000 sets. For prediction explainability, attention attribution scores were computed across 100 different anomalous sets with the *self-attention attribution* (Hao et al., 2021) method using 20 integration steps on all of the transformer layers. The models were provided the full sets for this analysis. For ISAB layers, we only utilized the first inner MAB/MSAB layer to compute the scores. The attribution scores were clipped to the range of $[0, \infty)$ and then aggregated across heads, layers, and rows via summation. Finally, the scores were normalized to the range of $[0, 1]$. This yields a single vector for a set of inputs where the value of each element indicates the importance of the corresponding input element.

Table A.13: The maximum-value regression architecture for UMBC.

| Output Size | Layer | Amount |
|---|---|---|
| $N_i \times 1$ | Input | $\times 1$ |
| $N_i \times 64$ | Linear(1, 64) | $\times 1$ |
| $4 \times 64$ | $SSE_4$ | $\times 1$ |
| $4 \times 64$ | SAB | $\times 1$ |
| $1 \times 64$ | $PMA_1$ | $\times 1$ |
| $1 \times 1$ | Linear(32, 1) | $\times 1$ |

Eyeglasses & Arched Eyebrows

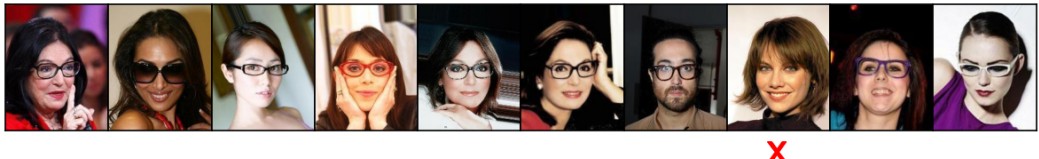

Mustache & Wearing Hat

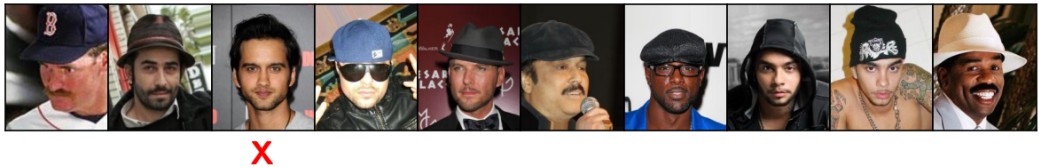

Figure A.4: Example sets containing an anomalous image.

Table A.14: The set anomaly classification architecture for ST, STrXL, and UST. Set Encoder corresponds to SAB for ST (SAB), $ISAB_4$ for ST (ISAB) and STrXL, and $U\text{-}ISAB_4$ for UST.

| Output Size | Layer | Amount |
|---|---|---|
| $N_i \times 768$ | Input | $\times 1$ |
| $N_i \times 32$ | Linear(768, 32) | $\times 1$ |
| $N_i \times 32$ | Set Encoder | $\times 4$ |
| $1 \times 32$ | $PMA_1$ | $\times 1$ |
| $1 \times 1$ | Linear(32, 1), Sigmoid | $\times 1$ |

Table A.15: The set anomaly classification architecture for UMBC.

| Output Size | Layer | Amount |
|---|---|---|
| $N_i \times 768$ | Input | $\times 1$ |
| $N_i \times 32$ | Linear(768, 32) | $\times 1$ |
| $4 \times 32$ | $SSE_4$ | $\times 1$ |
| $4 \times 32$ | SAB | $\times 3$ |
| $1 \times 32$ | $PMA_1$ | $\times 1$ |
| $1 \times 1$ | Linear(32, 1), Sigmoid | $\times 1$ |

