# OpenReview forum: "Universal Set Transformer: A Scalable and Interpretable Set/Multiset Architecture"
_ICLR.cc/2026/Conference — Submitted to ICLR 2026_

### Official Review · Reviewer_pUVJ · 2025-10-22

[review text omitted: it was posted to a different submission]

---

> ### Author Response · Authors · 2025-11-12
> **Incorrect Submission Review (Intended for Submission #3243)**
>
> It looks like this review was intended for submission #3243. This reviewer's review for our submission is visible in submission #3243: https://openreview.net/forum?id=JQR8OCptcG&noteId=gCeJeDsDkS

---

> ### Author Response · Authors · 2025-11-26
>
> We'd like to thank the reviewer for taking the time to review our work and for their helpful feedback.
>
> We agree that the discussion section did not cover all of the important limitations of the UST architecture. We have added two additional paragraphs (paragraphs 3 and 4) to the discussion section covering the multiset attention mechanism and its limitations. We believe that the discussion section now covers the primary limitations and tradeoffs of using our architecture over others.
>
> **1. What is the difference between the multiset attention as presented in this paper, compared to ref [1]?**
>
> Multiset Attention as proposed in our paper is unrelated to [1]. Multiset Attention is an optimization for multisets (sets containing exact duplicates) that allows factoring in element multiplicities (i.e. the number of times each element occurs) in the set to prevent redundant processing. Essentially, Multiset Attention enables sparse multiset processing at the cost of being strictly set-equivariant.
>
> The method proposed by [1] does not account for element multiplicities and the dense set containing duplicate elements must be processed in order for it to achieve multiset-equivariant processing (i.e. transforming each duplicate element distinctly), resulting in different final multiplicities. We have updated the first paragraph in Section 3.1 to describe the differences in set-equivariant and multiset-equivariant processing citing [1] and explain the necessity of set-equivariant processing for Multiset Attention.
>
> **References**
>
> 1. Zhang et al. Unlocking Slot Attention by Changing Optimal Transport Costs. ICML 2023.

---

> > ### Comment · Reviewer_pUVJ · 2025-11-27
> > **reply to rebuttal**
> >
> > Dear authors, thank you for the clarifications and paper improvements, I am increasing my score accordingly.

---

> ### Author Response · Authors · 2025-11-29
> **Incorrect Review After Revert**
>
> This review was incorrectly reverted to the other paper's review as mentioned previously in the initial comments above.

---

### Official Review · Reviewer_Ykyq · 2025-10-27

**Soundness:** 3
**Presentation:** 3
**Contribution:** 3
**Rating:** 4
**Confidence:** 3

**Summary:**

The authors propose a minibatch consistent permutation invariant model based on the set transformer. The model can produce elementwise representations of set while previous methods needed to use an invariant pooling mechanism which condenses the elementwise representations into a single vector or set of vectors.

**Strengths:**

- The proposed method builds upon the weaknesses of prior works which depend on a pooled set representation.

- The proposed method makes use of a flash-attention-like algorithm in the attention implementation to perform full attention while keeping the MBC property.

**Weaknesses:**

- L236 states: "We also found the unbiased gradient correction technique proposed by Willette et al. (2023) ineffective in our models." --> The referenced unbiased gradient technique was specifically derived for a model which has only elementwise transformations before an invariant pooling layer. This is the case with DeepSets, MBC, and UMBC. The proposed model uses a completely different architecture with transformer layers, so it is misleading to call the cited method ineffective, as it was not designed for the proposed setup.

-  L257: How can a minibatch containing only a few points perform worse? Is it only because the gradient contains a few points? If this is the case, it should only affect training and not inference, right?

**Questions:**

- Was UMBC in figure 1 (right) and table 3 trained with the full unbiased approxiamtion scheme?

Other than the Guassian clustering experiment which used DBSCAN to identify multisets by clustering close points, what other algorithms are used to identify multisets? I would be interested to see a latency tradeoff of scanning for multisets and performing the multiset attention vs. just doing regular attention without scanning for multisets.

For instance, if the set size is very large and there is only on repeat, then the scan would probably take longer than just processing the set normally with repeats. At what rate of repeat elements, does multiset processing show lower latency?

---

My main concern regards whether or not the full unbiased gradient approximation was used in the training of the UMBC baseline as well as the ablation study of the multiset latency.

---

> ### Author Response · Authors · 2025-11-26
>
> We'd like to thank the reviewer for reviewing our work and providing critical feedback.
>
> **1. L236: It is misleading to call the cited method ineffective, as it was not designed for the proposed setup**
>
> Unbiased gradient estimation works by accounting for missing gradients in mini-batches during pooling operations. The first transformer block within the U-ISAB is a permutation-invariant pooling of the input multiset, and the second transformer block within it is a permutation-equivariant un-pooling using cross attention with the input set and the pooled set. Since gradients are only present for a subset of those input elements (which are maintained on the U-ISAB layer outputs), we believe that we can apply the same unbiased gradient estimation method technique. The primary difference is that in the DeepSets, MBC, and UMBC models, there would only be one permutation-invariant pooling layer; while, in our approach, we would have multiple pooling layers. Therefore, we argue that this method is still applicable to our architecture if applied correctly. However, the additional U-ISABs operating on the full set are enough to overcome the biased gradient estimation, and we therefore do not notice any notable performance difference with or without unbiased gradient estimation with our architecture. We have adjusted our claim to state that we found that it was not effective for our UST architecture specifically. We have also added an additional sentence expanding on this statement indicating that it could be due to incompatibility with our architecture or that the additional full-set operations seem to overcome the biased gradient limitations.
>
> **2. L257: How can a minibatch containing only a few points perform worse?**
>
> For MBC models, the mini-batch size does not affect the result of inference; however it does not directly affect training either. The gradients computed between full-set and mini-batched batches are identical regardless of the mini-batch size. Therefore, it is the limiting of the number of elements which have gradients that can affect the training of the model. The gradients computed between full-set and mini-batched batches are identical regardless of the mini-batch size.
>
> For non-MBC models, the mini-batch size directly affects the result of inference and training. Mini-batches containing only a few points suffer from context fragmentation, making it more difficult for the models to make reliable predictions. The smaller the mini-batch, the less representative of the set the mini-batch becomes. Combining this with limited element gradients, non-MBC models have much more difficulty in training settings.
>
> We have updated the first paragraph in the MNIST point cloud section to clarify that it is non-MBC models specifically that are affected. We have also updated the following paragraph to explicitly discuss the performance differences in mini-batch and non-mini-batch training and how the gradient limitations affect the results.
>
> **3. Was UMBC in Figure 1 (right) and Table 3 trained with the full unbiased approximation scheme?**
>
> Yes, all UMBC models in our work were trained with the full unbiased gradient approximation. We have updated the first paragraph in the experiments section to clarify this detail.
>
> **4. Other than the Gaussian clustering experiment which used DBSCAN to identify multisets by clustering close points, what other algorithms are used to identify multisets? At what rate of repeat elements, does multiset processing show lower latency?**
>
> For identifying true multisets, one could use sorting or hash-based approaches to identify duplicate elements. For approximating multisets, clustering-based algorithms such as DBSCAN or voxelization could be used. We can determine what the multiset redundancy rate for any given multiset needs to be in order to deem a particular multiset identification/approximation algorithm to be beneficial over operating on the dense multiset. Given the $O(nk)$ baseline complexity of the U-ISAB with multiset attention, and the computational complexity of a particular multiset identification/approximation algorithm $O(C_n)$, the pre-processing algorithm is beneficial when $O(C_n + n(1-r)k) < O(nk) => r > C_n/(nk)$ where $n$ is the total set size (with duplicates), $k$ is the number of slots, and $r$ is the redundancy rate (percentage of elements that are duplicates) of the multiset. We have added a paragraph to the discussion section describing multiset identification/approximation methods as well as this analysis of when they may be beneficial.

---

> > ### Comment · Reviewer_Ykyq · 2025-11-27
> >
> > Thank you for your response. I have some clarifications to add based on my understanding of the UMBC framework and the unbiased gradient estimation technique.
> >
> > The unbiased gradient estimation of UMBC only holds as originally derived when $f(X)$ is sum decomposable. This is only the case if your queries are learned parameters and the keys and values are the set of interest, which is the case in a UMBC layer (or PMA in set transformer).
> >
> > This setup is only true for the first block of the proposed U-ISAB, and not the second block which switches the arguments and puts the input set as the queries and the pooled set from the first block as they keys and values. The output of this second block of U-ISAB could only be made unbiased under the original interpretation if you are considering the gradient w.r.t. the second argument. If you are storing gradient information based on indices of the complete U-ISAB outputs (i.e. the original set indices), then I believe this interpretation is invalid and must be fully re-derived for your setup.

---

> > > ### Author Response · Authors · 2025-12-02
> > >
> > > We’d like to thank the reviewer for pointing out this detail.
> > >
> > > The reviewer is correct in that the unbiased gradient estimation scheme proposed by [1] is not directly compatible with our architecture, and a new scheme would need to be derived. This realization does not affect any experiments/results in the main manuscript and only affects one result in the appendix of the manuscript which was used to show empirically that the scheme does not work with our architecture. We are unable to derive a new scheme and run experiments for our architecture within the constrained timeline. However, we have updated the manuscript to 1) exclude this one result in Section A.3.1 of the appendix and 2) correct our claim regarding ineffectiveness and instead explicitly indicate that the unbiased gradient estimation scheme by [1] is not compatible with our architecture.
> > >
> > > **References**
> > > 1. Wilette et al. Scalable Set Encoding with Universal Mini-Batch Consistency and Unbiased Full Set Gradient Approximation. ICML 2023.

---

### Official Review · Reviewer_ajaB · 2025-11-06

**Soundness:** 3
**Presentation:** 2
**Contribution:** 2
**Rating:** 4
**Confidence:** 3

**Summary:**

The paper proposes Universal Set Transformer (UST) which is a variant of the Transformer architecture that is specific for sets. The primary characteristics of the model are: 1. it's mini-batch consistent (MBC), meaning that different partitions of the set can be processed sequentially as "mini-batches" to reduce the peak memory requirements, and 2. it's efficient for multisets by removing repetitions and adding a multiplicity to each element in. On MNIST point could classification UST matches previous methods in the full set setting and outperforms in the mini-batch setting. Furthermore, UST shows favorable performance on both a synthetic clustering and bioinformatics taxonomy task.

**Strengths:**

- the UST is a clear improvement over previous set based neural network architectures for large input sets and multisets
- supporting multisets by adding the multiplicity is a neat way of supporting repeated elements in MSA

**Weaknesses:**

- one of the main points that the paper focuses is the memory footprint for large sets for (cross-)attention with O(kn). but papers like "Self-attention Does Not Need O(n^2) Memory" by Rabe and Staats show that this can be avoided by a more efficient implementation.
- the multiplicity is computed only for the input multiset. in later layers elements can still become more similar and collapse to "effectively equal" elements, hence the proposed architecture does not handle multisets in all generality

**Questions:**

1. It would be helpful to contextualize the work within the broader literature of efficient Transformer implementations. Does something like Rabe and Staats, 2021, eliminate the need for set-specific architecture designs for memory efficiency?
2. How do you compute multiplicity for similar but not equal elements? Or do elements count as repitions only when they are exactly equal?

---

> ### Author Response · Authors · 2025-11-26
>
> We'd like to thank the reviewer for their helpful feedback.
>
> **1. Does something like [1] eliminate the need for set-specific architecture designs for memory efficiency?**
>
> No. The work in [1] proposes the same softmax trick that we incorporate into our attention mechanism to process the input in chunks while producing identical output to processing the full input (i.e. MBC processing).This trick was seemingly rediscovered independently by [2] which is the paper we cite and build our architecture upon. While [1] could theoretically be used to address the memory concerns for processing sets, it does not address the $O(n^2)$ time complexity of self attention which still significantly limits scalability. Therefore, there is still good reason to seek out more efficient set-specific architectures. Our architecture leverages the softmax trick in [1, 2] and induced attention proposed by [3] to approximate full self-attention in $O(nk)$ time complexity while being MBC. Furthermore, our proposed multiset attention mechanism further reduces redundant computation. We have updated the manuscript to include [1] in our literature review in the third paragraph of the Introduction section.
>
> **2. How do you compute multiplicity for similar but not equal elements? Or do elements count as repetitions only when they are exactly equal?**
>
> For our proposed multiset attention, we only declare elements to be duplicates if they are exactly equal which can be computed using sorting or hash-based approaches. However, in some scenarios, it may be beneficial to interpret similar elements as the same. In these cases, one may need clustering based approaches like DBSCAN used in this work or voxelization. We added a paragraph (Paragraph 3) to the discussion section describing methodologies for computing element multiplicities and when it is beneficial to do so over operating on the dense multiset.
>
> **References**
>
> 1. Rabe and Staats. Self-attention Does Not Need $O(n^2)$ Memory. arXiv preprint 2021.
> 2. Wilette et al. Scalable Set Encoding with Universal Mini-Batch Consistency and Unbiased Full Set Gradient Approximation. ICML 2023.
> 3. Lee et al. Set Transformer: A Framework for Attention-based Permutation-Invariant Neural Networks. ICML 2019.

---

### Official Review · Reviewer_VhCM · 2025-11-08

**Soundness:** 3
**Presentation:** 3
**Contribution:** 3
**Rating:** 6
**Confidence:** 4

**Summary:**

- The Universal Set Transformer (UST) is a new architecture that enables scalable and interpretable transformer-based modeling of sets and multisets.
- It introduces a mathematically consistent mini-batch processing framework (MBC) that ensures identical results whether a set is processed all at once or in shards.
- UST adds Multiset Attention (MSA), which efficiently handles duplicate elements and reduces computational cost while maintaining full expressivity.
- Experiments show that UST matches or exceeds prior models' accuracy while using less memory and scaling effectively to very large sets.

**Strengths:**

- The paper is well-written, clearly structured, and effectively explains complex concepts like mini-batch consistency and multiset attention.
- The paper introduces the first transformer architecture that achieves true mini-batch consistency while preserving full self-attention expressivity.
- The architecture maintains attention-based interpretability while achieving strong accuracy and scalability across diverse tasks.

**Weaknesses:**

- UST still requires storing a full set instance before pooling, which limits true constant-memory scalability.
- The experimental evaluation is mainly limited to controlled benchmark and bioinformatics datasets, making it unclear how the model would perform on large-scale, real-world applications.
- The paper lacks a detailed analysis of interpretability, providing limited evidence on how attention scores meaningfully explain model predictions.
- Comparisons with more recent efficient transformer variants or non-attention-based set models are missing.

**Questions:**

- Are the gradients in UST theoretically equivalent to those obtained from full-set training, or only approximately consistent?
- Is there any quantitative or qualitative evidence that attention scores in UST provide meaningful interpretability?
- How does UST perform when applied to variable-sized sets with very different cardinalities across samples?
- Can the proposed framework be combined with sparse attention methods to further reduce computational cost?
- Some citations look strange to me. For example, "Deep Sets (Zaheer et al. (2017)) and FSPool (Zhang et al. (2020))" should be "Deep Sets (Zaheer et al., 2017) and FSPool (Zhang et al., 2020)."
- In Line 256, please check grammar for "Mini-batches containing few points cause these models suffer from context fragmentation"

**Details Of Ethics Concerns:**

There is no particular ethics concern.

---

> ### Author Response · Authors · 2025-11-26
>
> We'd like to thank the reviewer for their critical yet helpful feedback.
>
> **1. Are the gradients in UST theoretically equivalent to those obtained from full-set training, or only approximately consistent?**
>
> Yes, the gradients are theoretically equivalent to those in full-set and mini-batch training so long as all mini-batches have gradients associated with them. We have updated the first paragraph in Section 2.1.1 to indicate that this is the case.
>
> **2. Is there any quantitative or qualitative evidence that attention scores in UST provide meaningful interpretability?**
>
> We analyzed the attention scores resulting from UST and the other models for the Max-value regression task (Section 4.4.1) and Set Anomaly Detection task (Section 4.4.2) using the Self-Attention Attribution method [1]. We show in Figure 3 that UST performs comparable to Set Transformer (SAB and ISAB) and STrXL for identifying the max-value and anomaly in the sets for the tasks with full-set access. In the same figure, after mini-batched training with single-element mini-batches, we show that scores produced from UST identify the set anomaly with significantly more accuracy compared to the other models. In these results, we also show that previous MBC methods are much more limited in their interpretability. We have added a paragraph to the end of Section 4.4.2 summarizing the quantitative evidence of UST’s meaningful interpretability that was observed from the two experiments.
>
> **3. How does UST perform when applied to variable-sized sets with very different cardinalities across samples?**
>
> In order to explore the idea of variable-sized sets, we ran an additional experiment on the MNIST point cloud classification task dynamically varying the set sizes during training and testing. Due to the comment length restrictions here on OpenReview, the results will be shown in a follow-up post below.
>
> **4. Can the proposed framework be combined with sparse attention methods to further reduce computational cost?**
>
> Sparse attention methods [2] would break the permutation-equivariant property. However, STrXL has shown that approximately-permutation-equivariant models can still be effective [3], thus, it is reasonable to believe that sparse attention methods could be used. However, sparse attention is not directly compatible with multiset attention as sparse attention could result in the removal of interactions from all instances of elements that happen to have duplicates present. In normal dense set representations, sparse attention may remove interactions from some of the duplicate-element instances, but not all. Therefore, there would be a tradeoff. For situations where you are not working with multisets, sparse attention could be utilized for additional memory savings at the cost of breaking permutation-equivariance. We have added a paragraph (paragraph 4) to the discussion section of the manuscript covering the use cases of other attention-based memory reduction techniques when data has few-to-no duplicates.
>
> **5. Some citations look strange to me. For example, "Deep Sets (Zaheer et al. (2017)) and FSPool (Zhang et al. (2020))" should be "Deep Sets (Zaheer et al., 2017) and FSPool (Zhang et al., 2020).**
>
> We have updated the citations throughout the manuscript to now be formatted correctly as the reviewer has described.
>
> **6. In Line 256, please check grammar for "Mini-batches containing few points cause these models suffer from context fragmentation"**
>
> We have updated this line to correct this grammatical error.
>
> **References**
>
> 1. Hao et al. Self-Attention Attribution: Interpreting Information Interactions Inside Transformer. AAAI 2021.
> 2. Child et al. Generating long sequences with sparse transformers. arXiv preprint 2019.
> 3. Givens et al. STrXL: Approximating Permutation Invariance/Equivariance to Model Arbitrary Cardinality Sets. FLAIRS 2024.

---

> ### Author Response · Authors · 2025-11-26
>
> **Question #3 Continued**
>
> In order to explore the idea of variable-sized sets, we ran an additional experiment on the MNIST point cloud classification task dynamically varying the set sizes during training and testing. Each example is generated by first sampling a cardinality n uniformly from [32-512] for training and [32-1024] for testing, and then n points are sampled to form the point cloud following the same procedure described in the manuscript.
>
> In the table below, the models were trained with full access to the set without mini-batching and later evaluated with mini-batching varying the mini-batch size. UST outperforms the full-set evaluations here since the non-MBC models are now required to aggregate multiple mini-batches having never done so before, causing a significant performance reduction.
>
> | **Mini-batch Size** | **ST (SAB)**     | **ST (ISAB)**    | **STrXL**        | **UMBC**     | **UST**          |
> |-------------|------------------|------------------|------------------|--------------|------------------|
> | 1           | 11.48 ± 0.60     | 11.55 ± 0.70     | 23.44 ± 2.80     | 57.64 ± 4.11 | **92.54 ± 0.52** |
> | 2           | 12.10 ± 0.82     | 13.14 ± 1.58     | 29.39 ± 3.83     | 57.64 ± 4.11 | **92.54 ± 0.52** |
> | 4           | 16.42 ± 1.91     | 18.24 ± 2.04     | 37.18 ± 4.15     | 57.64 ± 4.11 | **92.54 ± 0.52** |
> | 8           | 29.94 ± 4.07     | 28.44 ± 1.87     | 45.77 ± 4.08     | 57.65 ± 4.11 | **92.54 ± 0.52** |
> | 16          | 43.07 ± 4.53     | 39.93 ± 1.72     | 52.98 ± 4.08     | 57.65 ± 4.11 | **92.54 ± 0.52** |
> | 32          | 51.88 ± 4.69     | 47.60 ± 1.28     | 57.03 ± 4.14     | 57.65 ± 4.11 | **92.54 ± 0.52** |
> | 64          | 56.98 ± 4.86     | 51.70 ± 1.23     | 59.66 ± 4.11     | 57.65 ± 4.11 | **92.54 ± 0.52** |
> | 128         | 61.05 ± 4.98     | 55.40 ± 1.25     | 62.59 ± 4.12     | 57.65 ± 4.11 | **92.54 ± 0.52** |
> | 256         | 66.39 ± 4.96     | 60.89 ± 1.59     | 67.52 ± 3.80     | 57.65 ± 4.11 | **92.54 ± 0.52** |
> | 512         | 75.74 ± 5.52     | 68.49 ± 2.73     | 76.13 ± 4.26     | 57.64 ± 4.11 | **92.54 ± 0.52** |
> | 1024        | **92.39 ± 0.44** | **92.62 ± 0.56** | **92.59 ± 0.61** | 57.65 ± 4.11 | **92.54 ± 0.52** |
>
> In the next table below, the models were trained with a constant mini-batch size of 8 (STrXL with a constant memory size of 8) and later evaluated with mini-batching varying the mini-batch size. Here, we observe that the MBC models outperform the standard Set Transformer models across the board for this task, while only outperforming STrXL once the mini-batch size becomes large enough. The memory mechanism within STrXL effectively scales each mini-batch which allows it to gain additional performance over the other models in the short run when few points were available in the mini-batches.
>
> | **Mini-batch Size** | **ST (SAB)** | **ST (ISAB)** | **STrXL**        | **UMBC**     | **UST**          |
> |-------------|--------------|---------------|------------------|--------------|------------------|
> | 1           | 16.50 ± 2.69 | 19.17 ± 3.89  | **76.85 ± 2.55** | 77.16 ± 2.06 | **80.28 ± 0.94** |
> | 2           | 40.52 ± 4.78 | 41.88 ± 5.16  | **82.68 ± 1.92** | 77.16 ± 2.06 | **80.28 ± 0.94** |
> | 4           | 69.85 ± 2.98 | 71.20 ± 3.97  | **87.13 ± 0.87** | 77.16 ± 2.06 | 80.28 ± 0.94     |
> | 8           | 80.17 ± 1.67 | 81.12 ± 1.77  | **88.86 ± 0.81** | 77.16 ± 2.06 | 80.28 ± 0.94     |
> | 16          | 80.79 ± 1.32 | 80.21 ± 2.08  | **88.51 ± 1.22** | 77.16 ± 2.06 | 80.28 ± 0.94     |
> | 32          | 78.93 ± 1.39 | 76.97 ± 2.38  | **87.09 ± 1.72** | 77.16 ± 2.06 | 80.28 ± 0.94     |
> | 64          | 75.97 ± 1.49 | 73.17 ± 2.59  | **85.26 ± 2.19** | 77.16 ± 2.06 | 80.28 ± 0.94     |
> | 128         | 71.26 ± 1.72 | 68.27 ± 2.64  | **82.70 ± 2.45** | 77.16 ± 2.06 | **80.28 ± 0.94** |
> | 256         | 62.83 ± 2.44 | 60.06 ± 2.68  | **78.40 ± 2.53** | 77.16 ± 2.06 | **80.28 ± 0.94** |
> | 512         | 50.70 ± 3.04 | 49.14 ± 3.11  | 69.68 ± 2.71     | 77.16 ± 2.06 | **80.28 ± 0.94** |
> | 1024        | 38.72 ± 3.82 | 38.91 ± 3.76  | 54.57 ± 4.12     | 77.16 ± 2.06 | **80.28 ± 0.94** |
>
> These results indicate that UST still outperforms other models even in variable cardinality scenarios. We added these evaluations to the appendix of the manuscript in Section A.3.1 and added a referral to these evaluations in the main text in the first paragraph of Section 4.1.

---

> > ### Comment · Reviewer_VhCM · 2025-11-27
> >
> > Thank you for the authors' rebuttal.
> >
> > I acknowledge I have read the rebuttal.
> >
> > I would maintain my score.

---

### Author Response · Authors · 2025-12-02
**Official Rebuttal Summary**

We would like to thank the reviewers again for their constructive feedback. We have revised the manuscript to fully address the questions and concerns they have raised by including the following changes:

1. To further demonstrate robustness, we added an experiment exploring variable set cardinalities during training and evaluation of the MNIST point cloud classification task to the appendix (Section A.3.1), further demonstrating UST’s performance over other models.
2. We extended the results discussion of Section 4.4.2 to provide a more detailed breakdown of the quantitative evidence of interpretability for UST and other baseline models.
3. We revised sections 2.1.1 and 3.1 to explicitly clarify that mini-batch training gradients are theoretically equivalent to full-set training and that UMBC models utilize unbiased gradient estimation.
4. We have expanded the literature review to include comparisons to other memory saving/multiset methods, highlighting the novelty and necessity of UST.
5. We expanded the discussion section to further cover UST’s limitations, practical guidelines detailing when pre-processing for multiset attention is advantageous, and compatibility with other computation-saving techniques (e.g. sparse attention).
6. Aligning with the established literature, we updated the manuscript to explicitly state that the unbiased gradient estimation scheme from [1] is theoretically incompatible with UST, and remove our experimental result from the appendix (Section A.3.1) that demonstrates this fact empirically.

**References**
1. Wilette et al. Scalable Set Encoding with Universal Mini-Batch Consistency and Unbiased Full Set Gradient Approximation. ICML 2023.

---

> ### Author Response · Authors · 2025-12-02
> **Important Note to ACs**
>
> The review by [pUVJ](https://openreview.net/forum?id=gIpuW5Ekiw&noteId=ZKub4lF0q1) currently shown is not for our paper.

---

### Meta-Review · Area_Chair_FFmR · 2026-01-06

**Summary:**

This paper proposes the Universal Set Transformer (UST), an extension of the Set Transformer (ST)—an attention-based, permutation-invariant architecture for set-structured inputs. A key limitation of the original ST is its poor scalability to very large sets that do not fit in memory: when such sets must be processed by sharding into mini-batches, the model’s output generally changes because standard attention is not mini-batch consistent. Prior work introduced mini-batch consistent (MBC) set encoders, including Slot Set Encoders (SSE) and Universal MBC (UMBC), to mitigate this issue, but these approaches typically rely on pooled/bottlenecked representations. UST aims to improve this by building an MBC generalization of ST that can still compute cross-element interactions. The paper also introduces a multiset extension, using a multiplicity-aware attention formulation that exploits duplicate elements to reduce redundant computation without materially increasing complexity.

**Reviewer Concerns:**

Reviewer VhCM
- UST requires storing full set before pooling: Not resolved. The authors admit this limitation in the main text - “While UST achieves state-of-the-art performance, it is important to note that, unlike UMBC where mini-batches are immediately discarded after use, the U-ISAB must maintain one full instance of the set prior to pooling”.
- Limited experiments: Not resolved. The authors did not directly address this point.
- Lacking detailed analysis on interpretability: partially resolved. The authors supplemented analysis on attention attribution matrices, but a fundamental question on whether the attention attribution could be a proxy of interpretability remains.
- No comparisons to more recent transformer variants: Not resolved. The authors did not directly address this point.

Reviewer ajaB
- The memory footprint issue raised by the paper can be resolved with more efficient implementations without new algorithms: Resolved. The authors clarified that while more efficient implementations can reduce memory usage, they do not address the $O(n^2)$ time complexity of self-attention.
- The multiplicity factor is computed only once, so the model may lose it while going deeper: Not resolved. The authors did not address this point.

Reviewer Ykyq
- The claim that the unbiased gradient estimator proposed in the UMBC paper is not effective for UST is not accurate, as it is tailored to the UMBC architecture: Resolved. The authors acknowledge that the original statement was misleading and agree to correct it (and to clarify the incompatibility with their architecture).

Reviewer pUVJ
- The paper is not discussing the limitation of UST: Resolved. The authors added limitations in the discussion.
- An important reference for multiset attention is missing:  Resolved. The authors clarified that the reference pointed out by the reviewer is not directly relevant to the notion of multiset attention considered in this submission.

**Reviewer Scores:**

Overall, I find the paper to be borderline even after the rebuttal. While some concerns were resolved or partially addressed, several fundamental limitations remain.

The most significant limitation—explicitly acknowledged by the authors—is that the proposed model requires storing the dataset throughout the entire forward pass when stacking multiple U-ISAB layers. This contrasts with UMBC, which can discard the dataset once it has been projected onto slots. This design choice may constitute a serious bottleneck: not only does it require retaining large sets in memory, but it also deviates from the streaming setting, which is an important motivation behind the MBC.

Moreover, I find the claim that UST can compute cross-element interactions whereas UMBC cannot to be an overstatement. After projecting inputs to slots, UMBC can process these slots via self-attention, which allows information to flow between elements indirectly through the slot bottleneck. This point is also closely tied to the paper’s interpretability claim. By linking attention from data to slots and from slots to slots, one can in principle recover data-to-data attention attribution matrices, even if these attributions may be less direct or less pronounced than those obtained via explicit element-wise self-attention. Apart from the broader and still open question of whether attention attribution matrices can serve as a reliable proxy for interpretability, the stronger claim that “UMBC is not interpretable” is therefore not accurate.

For these reasons, I recommend rejection.

---

### Decision · Program_Chairs · 2026-01-26

Reject